# *Cyfip1* haploinsufficient rats show white matter changes, myelin thinning, abnormal oligodendrocytes and behavioural inflexibility

Ana I. Silva [1,2,3], Josephine E. Haddon[1,2,4], Yasir Ahmed Syed[1,5], Simon Trent[1,2], Tzu-Ching E. Lin[1,2], Yateen Patel[1,2], Jenny Carter[1,2], Niels Haan [1,2], Robert C. Honey[4], Trevor Humby[1,4], Yaniv Assaf[6], Michael J. Owen [1,2], David E.J. Linden[1,2,3,7], Jeremy Hall[1,2] & Lawrence S. Wilkinson[1,2,4]

The biological basis of the increased risk for psychiatric disorders seen in 15q11.2 copy number deletion is unknown. Previous work has shown disturbances in white matter tracts in human carriers of the deletion. Here, in a novel rat model, we recapitulated low dosage of the candidate risk gene *CYFIP1* present within the 15q11.2 interval. Using diffusion tensor imaging, we first showed extensive white matter changes in *Cyfip1* mutant rats, which were most pronounced in the corpus callosum and external capsule. Transmission electron microscopy showed that these changes were associated with thinning of the myelin sheath in the corpus callosum. Myelin thinning was independent of changes in axon number or diameter but was associated with effects on mature oligodendrocytes, including aberrant intracellular distribution of myelin basic protein. Finally, we demonstrated effects on cognitive phenotypes sensitive to both disruptions in myelin and callosal circuitry.

[1] Neuroscience and Mental Health Research Institute, MRC Centre for Neuropsychiatric Genetics and Genomics, Hadyn Ellis Building, Cathays, Cardiff CF24 4HQ, UK. [2] Division of Psychological Medicine and Clinical Neurosciences, School of Medicine, Cardiff University, Cardiff, CF24 4HQ, UK. [3] Cardiff University Brain Research Imaging Centre, School of Psychology, Cardiff University, Maindy Road, Cardiff CF24 4HQ, UK. [4] School of Psychology, Cardiff University, Cardiff, Park Place CF10 3AT, UK. [5] School of Bioscience, The Sir Martin Evans Building, Museum Ave, Cardiff CF10 3AX, UK. [6] Department of Neurobiology, Tel-Aviv University, Ramat Aviv, 6997801 Tel-Aviv, Israel. [7] School of Mental Health and Neuroscience, Maastricht University, Maastricht, 6229 ER, The Netherlands. Correspondence and requests for materials should be addressed to J.H. (email: Hallj10@cardiff.ac.uk) or to L.S.W. (email: wilkinsonl@cardiff.ac.uk)

Low gene dosage of the cytoplasmic FMRP interacting protein 1 (CYFIP1) gene is a candidate risk factor for psychopathology by virtue of its involvement in the pathogenic 15q11.2 BP1-BP2 copy number variant (CNV). Heterozygous deletion of this genomic interval leads to a two- to four fold increase in risk for intellectual disability and psychiatric problems, including schizophrenia, autism, as well as a significant increase in the risk for epilepsy[1,2]. The deletion contains four genes: non-imprinted in Prader-Willi/Angelman syndrome 1 gene (NIPA1), non-imprinted in Prader-Willi/Angelman syndrome 2 gene (NIPA2), CYFIP1 and tubulin gamma complex associated protein 5 gene (TUBGCP5)[3]. Whilst all these genes are expressed in the brain and may be of potential relevance to psychopathology, CYFIP1 haploinsufficiency is considered to be a likely significant contributor to the 15q11.2 BP1-BP2 psychiatric phenotype due to its known involvement in a number of key brain plasticity-related functions. These include alterations in dendritic spine morphology and branching, mediated by interactions in two distinct complexes: the WAVE regulatory complex to modulate ARP2/3 dependent actin cytoskeleton dynamics, and CYFIP1-eIF4E complex to suppress protein translation at the synapse through interactions with fragile X mental retardation 1 protein (FMRP), the gene product of FMR1[4]. Mutations in FMR1 are causative for fragile X syndrome, a condition associated with intellectual disability and a range of psychiatric symptoms[5].

Changes in white matter microstructure have been reported consistently in major psychiatric disorders including schizophrenia, autism and intellectual disability[6]. Moreover, using diffusion tensor imaging (DTI) methods, in recently published findings we found extensive white matter changes in 15q11.2 BP1-BP2 CNV carriers, specifically widespread increases in fractional anisotropy (FA) in deletion carriers[7]. Some of the biggest changes we observed were in the posterior limb of the internal capsule and corpus callosum. Prominent effects in the corpus callosum are consistent with previous findings by others of increased corpus callosum volume in 15q11.2 BP1-BP2 deletion subjects[8]. The human data raise three main questions; which of the four genes in the 15q11.2 BP1-BP2 interval are important for the disturbances in white matter microstructure, what are the cellular changes underlying the white matter effects, and what are the functional consequences of the white matter changes in the context of the increased risk for disorder. Given the potential major impact of CYFIP1 in 15q11.2 BP1-BP2 associated phenotypes, in the present work we addressed these questions by taking advantage of the enhanced experimental tractability of a Cyfip1 haploinsufficiency rat line (hereafter designated Cyfip1+/−) created using CRISPR/Cas9 technology modelling the reduced gene dosage of CYFIP1 in 15q11.2 BP1-BP2 deletion carriers.

Our focus on white matter microstructure was also guided by evidence that CYFIP1 is an actin regulator, and thus likely to affect white matter via the requirement of precise regulation of the actin cytoskeleton for normal cellular development, morphology and migration. Hence, CYFIP1 haploinsufficiency has the potential to disrupt axonal organisation via both effects on axonal guidance[9] and the myelin component of white matter tracts[10,11]. Myelin is produced by mature oligodendrocytes and several studies have linked actin regulators to oligodendrocyte-myelin dynamics. The Wiskott-Aldrich Syndrome protein family member 1 (WAVE1) and the integrin-linked kinase (ILK) regulate oligodendrocyte differentiation and axon ensheathment[12,13], while the Arp2/3 complex, a key actin nucleator, is required for initiation of myelination[11], and Rho GTPases Cdc42 and Rac1 regulate myelin sheath formation[14].

We therefore hypothesised there would be white matter abnormalities in the Cyfip1+/− rat line possibly linked to underlying changes in axonal architecture including myelin. We also anticipated functional effects on brain and behaviour on the basis that axon-myelin perturbations can have marked effects on brain network activity caused by disruptions in the temporal coherence of action potential integration across different brain regions[15,16]. Synchronisation of synaptic signals is crucial in learning, and a previous study in shiverer (deletion mutant of myelin basic protein (MBP)) and mld (allelic mutant to shiverer with lowered MBP expression) mice[15] showed that deficits in myelination had a specific effect on behavioural flexibility in a reversal learning task. In the present work therefore, we looked for evidence of maladaptive brain function in the Cyfip1+/− rats using behavioural tasks that assayed behavioural flexibility.

## Results

**Cyfip1 haploinsufficiency disrupts white matter microstructure.** Full details of the creation of the Cyfip1+/− rat model are in the Supplementary Methods. CRISPR/Cas9 targeting led to a 4 bp out of frame heterozygous deletion in exon 7 of the Cyfip1 gene at location Chromosome 1: 36974–36977 and a resulting bioinformatics prediction of an early stop codon in exon 8, which was verified functionally using qPCR and Western Blot to measure reductions in mRNA and protein respectively.

To investigate white matter microstructure in the Cyfip1+/− rat brain a cohort of 24 behaviourally naïve male rats (wild-type (WT) $n = 12$, Cyfip1+/− $n = 12$) were anaesthetised with isoflurane in oxygen at 4% and maintained at 1%, and DTI data were collected using a 9.4 T MRI scanner, utilising 60 noncollinear gradient directions with a single b-value shell at 1000 sm$^{-2}$. Group comparisons were carried out using Tract-Based Spatial Statistics (TBSS)[17] available in FMRIB Software Library (FSL), with a randomise function allowing voxel-wise nonparametric permutation analysis of the DTI maps projected onto a whole brain white matter skeleton (Supplementary Fig. 1). The randomise function was used with the threshold-free cluster enhancement (TFCE)[18], generating cluster-size statistics based on 1000 random permutations. Behaviourally naïve animals were used in the light of evidence showing behaviour itself can influence white matter[19,20]. Figure 1 shows the regions where significant differences in white matter microstructure were found after correction for multiple comparisons. Figure 1a shows the pattern of changes using a highly conservative family-wise error (FWE) correction. This approach showed consistent reductions in FA in the corpus callosum, in the external and internal capsule, and in parts of the fimbria/fornix in Cyfip1+/− rats, with no differences in axial diffusivity (AD), radial diffusivity (RD) and mean diffusivity (MD). We complemented the highly conservative FWE correction method used in human imaging studies with the False Discovery Rate (FDR) correction for multiple comparisons based on the Benjamini–Hochberg procedure[21], used previously in rodent imaging data[19,22]. This analysis, shown in Fig. 1b, revealed additional white matter changes including increases in FA in regions of the fornix and fimbria suggesting that Cyfip1 haploinsufficiency may have differential effects in different brain regions. Figure 1b also shows changes in other DTI metrics, after FDR correction, illustrating mostly decreases in AD and increases in RD, and MD. These effects were complementary in terms of (a) being localised in the corpus callosum and external and internal capsule and (b) being consistent with the overall predominant effects of Cyfip1 haploinsufficiency in reducing FA.

We next manually generated binary masks of regions of interest (corpus callosum, internal capsule, external capsule and fimbria/fornix), guided by the results from FWE correction using FSL (Supplementary Fig. 2), and assessed mean FA, AD, RD and MD in these white matter tracts. As can be seen in Table 1,

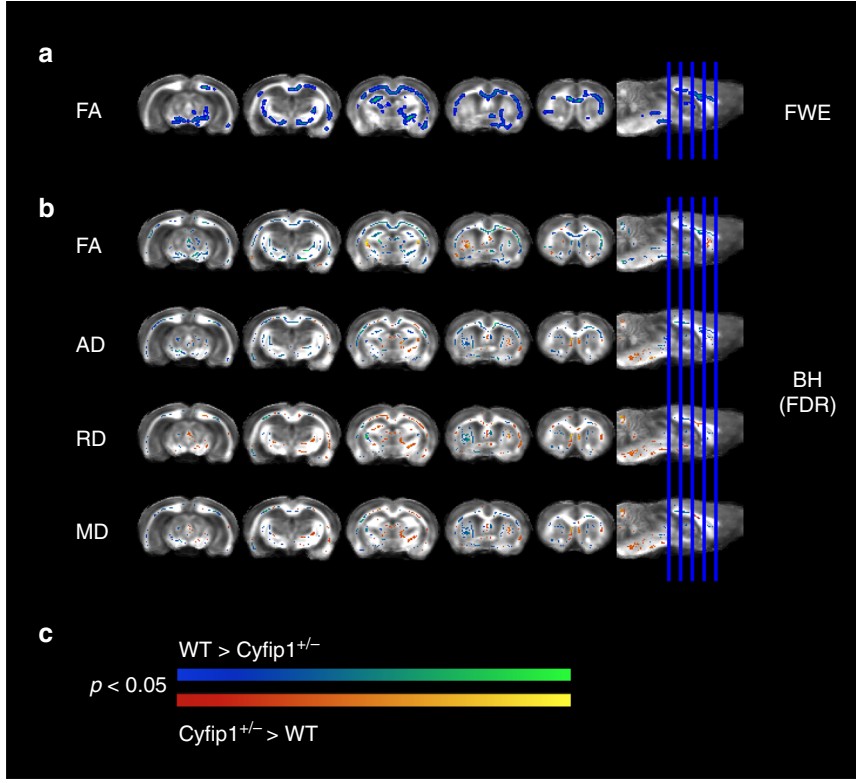

**Fig. 1** *Cyfip1* haploinsufficiency disrupts white matter microstructure. White matter changes comparing WT (*n* = 12) and *Cyfip1*[+/−] (*n* = 12) rats. Data shows significant group differences using two-tailed unpaired t-tests based on Threshold-Free Cluster Enhancement (TFCE) algorithm after **a** family-wise error (FWE) rate correction for fractional anisotropy (FA), and **b** false discovery rate (FDR) correction based on the Benjamini–Hochberg procedure for FA, axial diffusivity (AD), radial diffusivity (RD) and mean diffusivity (MD). All the parametric maps were generated at a significance level of *p* < 0.05. **c** Scale bars indicating the direction of the changes in both **a**, **b**, where relative decreases in *Cyfip1*[+/−] rats are represented by a gradient of blue (less significant) to green (more significant), and relative increases in *Cyfip1*[+/−] are represented by a gradient of red (less significant) to yellow (more significant)

**Table 1 Quantification of DTI changes in regions where significant differences in FA were seen in TBSS analysis after FWE correction**

| ROIs | FA | | AD ($10^{-2}$) | | RD ($10^{-3}$) | | MD ($10^{-3}$) | |
|---|---|---|---|---|---|---|---|---|
| | **WT** | **Cyfip1**[+/−] | **WT** | **Cyfip1**[+/−] | **WT** | **Cyfip1**[+/−] | **WT** | **Cyfip1**[+/−] |
| CC | 0.49 ± 0.02 | 0.46 ± 0.02[a] | 0.14 ± 0.005 | 0.13 ± 0.006[a] | 0.69 ± 0.05 | 0.71 ± 0.04 | 0.92 ± 0.04 | 0.91 ± 0.03 |
| IC | 0.45 ± 0.02 | 0.44 ± 0.02 | 0.11 ± 0.007 | 0.11 ± 0.007 | 0.53 ± 0.03 | 0.54 ± 0.03 | 0.74 ± 0.04 | 0.73 ± 0.04 |
| EC | 0.38 ± 0.02 | 0.36 ± 0.02[a] | 0.12 ± 0.004 | 0.12 ± 0.003 | 0.73 ± 0.03 | 0.73 ± 0.02 | 0.89 ± 0.03 | 0.88 ± 0.02 |
| FF | 0.50 ± 0.03 | 0.48 ± 0.02 | 0.17 ± 0.008 | 0.16 ± 0.004 | 0.69 ± 0.06 | 0.70 ± 0.04 | 1.01 ± 0.06 | 1.00 ± 0.03 |

Source data are provided as a Source Data file
*FA* fractional anisotropy, *AD* axial diffusivity, *RD* radial diffusivity, *MD* mean diffusivity values from WT and Cyfip1[+/−] rats, *ROIs* regions of interest, *CC* corpus callosum, *IC* internal capsule, *EC* external capsule, *FF* fornix/fimbria
Results obtained using TBSS-based ROI analysis, mean ± standard deviation, two-tailed unpaired *t*-test, [a] < 0.05

analysing the DTI data in this way (which averaged differences between WT and *Cyfip1*[+/−] rats within a discrete fibre tract, as opposed to the voxel-by-voxel analysis which detected clusters of voxel differences in white matter tracts across the whole brain) showed that the most significant differences were reductions in FA in the corpus callosum (*t* = 2.3, df = 20.75, *p* < 0.05) and external capsule (*t* = 2.4, df = 22, *p* < 0.05) in the *Cyfip1*[+/−] rats compared to WT, as assessed with a two-tailed unpaired *t*-test. These data were consistent with the previous voxel-by-voxel analysis and provided the additional finding that the most extensive white matter changes in the *Cyfip1*[+/−] rats occurred in these structures.

**Cyfip1 haploinsufficiency affects myelin in corpus callosum.**
We next investigated the cellular nature of the *Cyfip1* associated

DTI changes. DTI measures can be affected by several factors and previous studies have linked decreases in FA in white matter tracts with less myelin, lower axonal density, axonal damage, or changes in axonal organisation[23,24]. To assess cellular changes, we carried out an ultra-structural analysis, blind to genotype, using transmission electron microscopy in behaviourally naïve animals focusing on the corpus callosum, given the DTI data indicating the sensitivity of this structure to *Cyfip1* haploinsufficiency. The experiment used a new cohort of rats (WT *n* = 5, *Cyfip1*[+/−] *n* = 4). In order to obtain a representative sample, we sampled 15 regions across the anterior-posterior extent of the corpus callosum encompassing the genu, body and splenium, from sagittal brain sections (representative micrographs in Fig. 2a). We measured the number of myelinated and unmyelinated axons, the inner diameter and the outer diameter (including the myelin

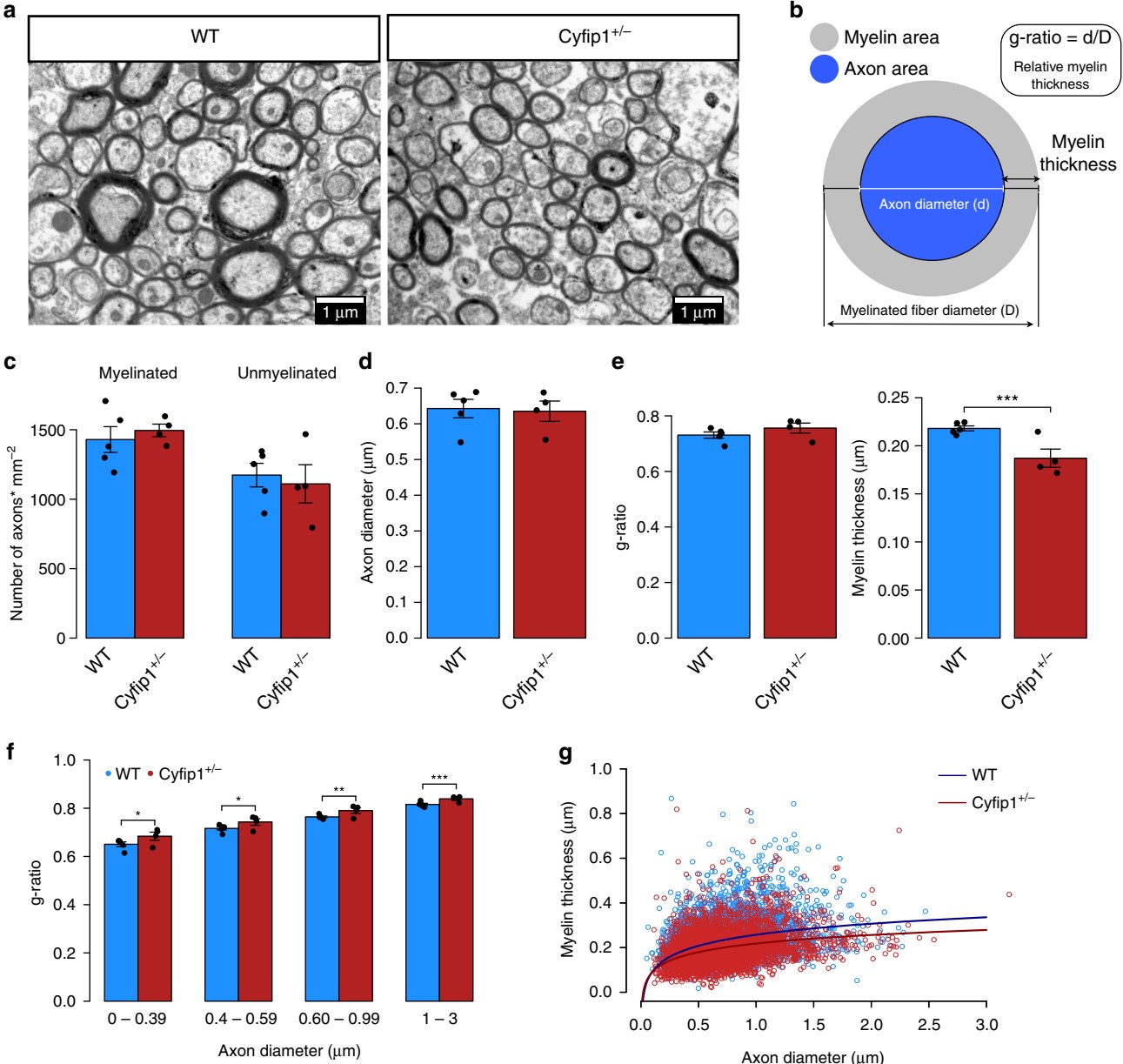

**Fig. 2** Decreased myelin thickness in the corpus callosum in the *Cyfip1*[+/−]rats. **a** Representative electron micrographs of axons in the WT (n = 5 animals and n = 7148 axons) and *Cyfip1*[+/−] (n = 4 animals and n = 5979 axons) rats. **b** Schematic illustration of the axon and myelin sheath and calculation of the g-ratio and myelin thickness. **c** Mean number of myelinated ($t = -0.63$, df = 5.72, $p = 0.55$, ns) and unmyelinated axons per group ($t = 0.39$, df = 5.15, $p = 0.71$, ns), assessed with a two-tailed unpaired $t$-test. **d** Mean axon diameter of myelinated fibres per group (LME: $\chi^2(1) = 0.05$, $p = 0.83$). **e** Mean g-ratio per group (LME: $\chi^2(1) = 2.03$, $p = 0.15$) and mean myelin thickness per group (LME: $\chi^2(1) = 14.63$,***), showing significant decreased myelin thickness in *Cyfip1*[+/−] rats. **f** mean g-ratios calculated for small (n = 1510 WT and 1276 *Cyfip1*[+/−] axons; LME: $\chi^2(1) = 4.23$,*), medium-small (n = 2283 WT and 2043 *Cyfip1*[+/−] axons; LME: $\chi^2(1) = 4.44$,*), medium-large (n = 2551 WT and 1993 *Cyfip1*[+/−] axons; LME: $\chi^2(1) = 7.14$,**), and large (n = 804 WT and 667 *Cyfip1*[+/−] axons; LME: $\chi^2(1) = 13.92$,***) myelinated axons, showing significant increases in g-ratio in all different axon diameter ranges, and more significant in larger axons. **g** Scatter plot of myelin thickness values across all axon diameters WT (n = 7148 axons) and *Cyfip1*[+/−] (n = 5979 axons). Differences between axon diameter, g-ratio and myelin thickness measures were assessed using linear mixed effects (LME) models adjusted for individual variability. Data are mean ± SEM; *<0.05, **<0.01, ***<0.001. Source data are provided as a Source Data file

sheath) of each axon, and then calculated the myelin thickness and the g-ratio (myelin thickness relative to axon diameter, where smaller g-ratios indicate greater myelin thickness) of each myelinated axon (see measures taken in Fig. 2b).

We used linear mixed effects (LME) models to analyse the effect of genotype on axon diameter, g-ratio and myelin thickness, considering variation across animals, whereas a two-tailed unpaired t-test was used to compare the number of axons

between groups. In this analysis, the myelin thickness was log-transformed since the data followed a log-normal distribution, whereas the other measures followed a normal distribution. No genotype differences were found in the number of unmyelinated ($t = 0.39$, df = 5.15, $p = 0.71$) and myelinated ($t = -0.63$, df = 5.72, $p = 0.55$) axons (Fig. 2c), or in the diameter of the axons (Fig. 2d, LME: $\chi^2(1) = 0.05$, $p = 0.83$), suggesting no differences in axonal density and calibre in the corpus callosum of

the $Cyfip1^{+/-}$ rats. The analyses did not show a significant increase in g-ratio when comparing all axons in each group (LME: $\chi^2(1) = 2.03$, $p = 0.15$), however it revealed a significant reduction in myelin thickness in the $Cyfip1^{+/-}$ rats (LME: $\chi^2(1) = 14.63$, $p < 0.001$), both shown in Fig. 2e. The fact that we did not see a significant difference in g-ratio could have resulted from variability in the average of axon diameters within animals in the same group, which is related to g-ratio (Supplementary Fig. 3). Furthermore, we needed to consider that the extent of myelination can be related to axon diameter[25], and whether the effects on the g-ratio were specific to certain sizes of axons. Analysing the g-ratio of axons within specific diameter ranges revealed a significant increased g-ratio in each interval in the $Cyfip1^{+/-}$ rats (Fig. 2f), that was more significant in larger axons. These analyses indicate decreased myelin thickness in the corpus callosum of the $Cyfip1^{+/-}$ rats that is more pronounced in larger axons.

**$Cyfip1^{+/-}$ rats have less oligodendrocytes in corpus callosum.** Myelin is produced by mature oligodendrocytes, so we next tested whether $Cyfip1$ haploinsufficiency influenced the number and/or the maturation of oligodendrocytes using antibodies to the specific molecular markers Olig2 and Cc1. This experiment used rats taken randomly from the same group of rats providing the DTI data shown in Fig. 1 (WT $n = 7$ WT and $Cyfip1^{+/-}$ $n = 7$). The analysis focused on the corpus callosum and external capsule and at least four random sections were taken for quantification in each rat from coronal sections. Sections were stained for Olig2 and Cc1 proteins. Cells stained for Olig2 alone represented all the oligodendrocyte lineages from early progenitors to mature cells, whereas cells double-stained for Olig2 and Cc1 proteins revealed specifically the mature oligodendrocyte (myelin-producing) population. In the $Cyfip1^{+/-}$ rats, this analysis showed a

significant reduction in both the number of oligodendrocyte lineage cells (Fig. 3a; $t = 2.18$, df = 11.94, $p < 0.05$) and mature oligodendrocytes ($t = 2.48$, df = 11.99, $p < 0.05$) in comparison with WT. We also found a significant reduction in the level of MBP (Fig. 3b, $t = 2.16$, df = 11.96, $p = 0.052$) in the corpus callosum/external capsule of the $Cyfip1^{+/-}$ rats. The significance was assessed using a two-tailed unpaired $t$-test.

**$Cyfip1^{+/-}$ rats have aberrant MBP distribution in oligodendrocytes.** We used primary cell culture methods to address further the question of how Cyfip1 haploinsufficiency might impact on oligodendrocytes and myelination. We used standard protocols[26], which generate oligodendrocyte precursor cells (OPC) at ≥95% purity from WT and $Cyfip1^{+/-}$ rats. Three independent biological replicates were performed. After 3 days of differentiation the cells were processed for immunohistochemistry with antibodies to O4 and MBP. Cells stained for O4 alone mark the early, immature stages of oligodendrocyte differentiation and combined O4/MBP staining mark later mature stages of oligodendrocyte differentiation. For the imaging analysis five images from random visual fields were taken per well and a minimum of 1000 cells quantified per experimental group/ replicate, as previously, we used a LME models for statistical analysis to account for variation across biological repeats (where these were considered random effects).

We focused on MBP, which is both a marker for mature oligodendrocyte differentiation and essential for the production of myelin[27]. With the enhanced cellular resolution possible with cultured oligodendrocytes, we immediately noticed that the staining pattern for MBP looked markedly different in cells originating from $Cyfip1^{+/-}$ brain tissue compared to WT. Specifically, as illustrated in Fig. 4a, there appeared to be a more punctate organisation in $Cyfip1^{+/-}$ cells where MBP staining was

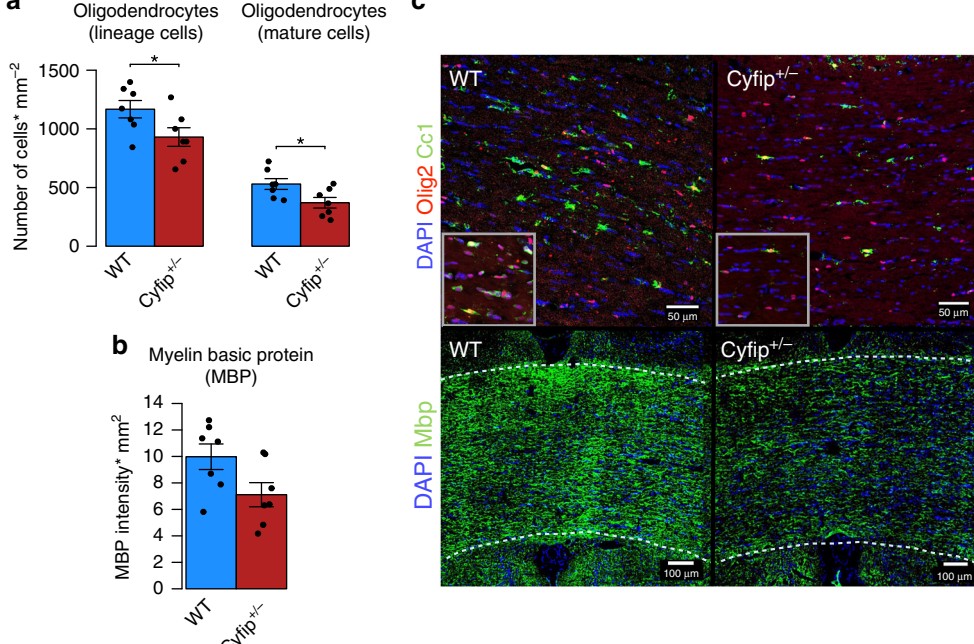

**Fig. 3** Altered number of oligodendrocyte lineage and mature cells, and levels of myelin basic protein (MBP), in the corpus callosum of the $Cyfip1^{+/-}$ rats. **a** Mean number of oligodendrocyte lineage ($n = 7$ each; $t = 2.18$, df = 11.94, *), stained with Olig2, and mature ($n = 7$ each; $t = 2.48$, df = 11.99, *), stained with Olig2 and Cc1, cells per mm2. **b** Mean MBP intensity multiplied by the percentage area (mm2) of the staining ($n = 7$ each; $t = 2.16$, df = 11.96, $p = 0.052$). **c** Representative images (at magnification ×20 (top) and ×10 (bottom)) for the following immunomarkers: DAPI, Olig2, Cc1 and Mbp in the corpus callosum of the WT and $Cyfip1^{+/-}$ rats. Scale bars = 50 μm (top) and 100 μm (bottom). Differences between groups were assessed using a two-tailed unpaired $t$-test. Data are mean ± SEM; *<0.05, **<0.01, ***<0.001. Source data are provided as a Source Data file

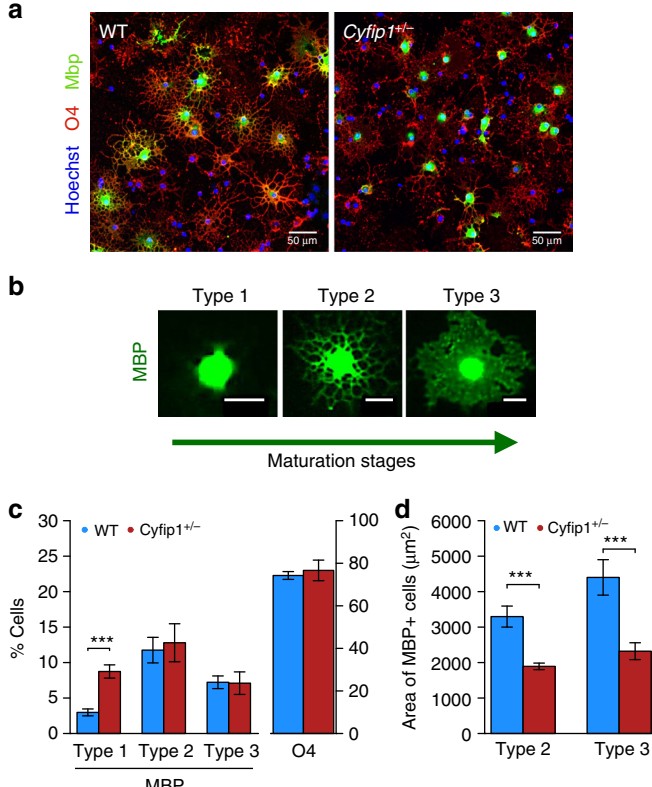

extending into the cell processes; Type 3, final maturation stage where MBP is throughout the cell and distributed within the membranous processes of cells to give rise to a 'spider's web-like' appearance (see Fig. 4b for representative examples). The quantitative data confirmed the previous qualitative observation in revealing a highly significant increased incidence of Type 1 cells in $Cyfip1^{+/-}$ (LME: $\chi^2(1) = 68.49$, $p < 0.001$, Fig. 4c). Furthermore, whilst there were no significant genotype differences in the overall proportion of cells classified, broadly, as Type 2 (LME: $\chi^2(1) = 0.59$, $p = 0.44$) or Type 3 (LME: $\chi^2(1) = 0.02$, $p = 0.88$, Fig. 4c), within each of these classifications $Cyfip1^{+/-}$ oligodendrocytes showed a more constrained cellular distribution of MBP as indicated by a highly significant and consistent reduction in the area of MBP staining in the $Cyfip1^{+/-}$ population, which amounted to about 50% of that seen in WT cells (LME: $\chi^2(1) = 258.03$, $p < 0.001$, Fig. 4d). The area of MBP staining observed in the WT cells was consistent with previous findings by others using similar culture methods[14]. Furthermore, the effects of $Cyfip1$ haploinsufficiency occurred in the absence of any observable genotype effects on MBP + oligodendrocyte cell size/gross morphology or the overall number of differentiating oligodendrocytes present in the cultures as evidenced by the total numbers of O4 + cells. (Fig. 4c, LME: $\chi^2(1) = 1.44$, $p = 0.23$). Together, these data were consistent with $Cyfip1$ haploinsufficiency hindering the translocation of MBP to the distal parts of the highly branched mature oligodendrocyte, a process that is critical for successful differentiation and the production of myelin[29], and likely therefore to be of relevance to the previous finding of myelin thinning in the $Cyfip1^{+/-}$ rats.

**Fig. 4** $Cyfip1$ haploinsufficiency influences the intracellular distribution of myelin basic protein (MBP) in cultured oligodendrocytes. **a** Immunostaining of oligodendrocytes for MBP and O4 from wild type (WT) and $Cyfip1^{+/-}$, scale bar = 50 μm, illustrating the punctate intracellular pattern of staining of MBP in $Cyfip1^{+/-}$ relative to the more diffused, widespread pattern of staining in WT. **b** Representative images of oligodendrocytes staining for MBP exhibiting features of Type 1 (MBP localised to the cell body), Type 2 (MBP ramifying into the cell processes), Type 3 (MBP distributed throughout the cell and within the membranes of cell processes) reflecting increasing maturation stages of oligodendrocytes, scale bar = 20 μm. **c** the effects of genotype on the percentage of Type 1 ($n = 172$ WT and 637 $Cyfip1^{+/-}$ cells; LME: $\chi^2(1) = 68.49$, ***), Type 2 ($n = 735$ WT and 905 $Cyfip1^{+/-}$ cells, LME: $\chi^2(1) = 0.59$, $p = 0.44$), Type 3 ($n = 456$ WT and 454 $Cyfip1^{+/-}$ cells; LME: $\chi^2(1) = 0.02$, $p = 0.88$), and O4 + cells ($n = 4917$ WT and 5789 $Cyfip1^{+/-}$ cells; LME: $\chi^2(1) = 1.44$, $p = 0.23$), as a proportion of all cells ($n = 6623$ WT and 7845 $Cyfip1^{+/-}$ cells) in the culture (stained with Hoechst); this panel also illustrates the effects of genotype on the overall proportion of differentiating oligodendrocytes as indexed by all cells staining for O4. **d** effects of genotype on the area of intracellular MBP staining in Type 2 ($n = 491$ WT and 591 $Cyfip1^{+/-}$ cells; LME: $\chi^2(1) = 258.03$, ***) and Type 3 ($n = 265$ WT and 341 $Cyfip1^{+/-}$ cells; LME: $\chi^2(1) = 145.52$, ***) oligodendrocytes, as depicted in the representative images in b. Values were obtained from 3 independent experiments. Differences between number of cells and area were quantified using linear mixed effects (LME) models adjusted for variability in each biological repeat. Data are mean ± SEM; *<0.05, **<0.01, ***<0.001. Source data are provided as a Source Data file

localised mainly to the cell body region, compared to a more widely distributed pattern of cellular staining in WT that extended into the cell processes. We interrogated this finding in more detail using a quantitative analysis that classified MBP + cells into three previously established categories indexing the maturation stages of oligodendrocytes that culminates in the formation of compact myelin[28]: Type 1, MBP localised only to the cell body; Type 2, any ramified MBP intracellular distribution

**$Cyfip1$ haploinsufficiency affects behavioural flexibility.** We next assessed whether the $Cyfip1$ related imaging and cellular phenotypes were associated with effects on behaviour. Behavioural changes have been observed in rodent models of reduced myelination including *shiverer* and *mld* mice with modifications in myelin basic protein[15]. Both mutants showed highly specific effects on behavioural flexibility in a reversal learning task, whereby they had difficulty changing their behaviour to reflect the new reward contingencies, without concomitant deficits in learning deficits *per se*[15]. Furthermore, both human[30–35] and animal studies[36,37] have also shown that disruptions to connectivity involving callosal circuitry and circuits involving the internal and external capsules impact on a number of psychological functions, in particular those mediating attention and response control, especially response inhibition.

To assay behavioural flexibility, we utilised a touch screen-based appetitive reversal learning task in a separate cohort of experimental rats (WT $n = 7$, $Cyfip1^{+/-}$ $n = 10$). The reversal learning task first allowed an assessment of basic appetitive learning where rats had to learn that one visual stimulus was associated with reward (the S+) and another stimulus was not (S−), with the two stimuli counterbalanced across animals. This was followed by reversal of the contingencies (see Supplementary Fig. 4 for flowchart of task design). Successful reversal learning can be aligned to tasks used in clinical studies (i.e. the Stroop task, SART, stop signal) to examine cognitive processes supporting flexible control of behaviour[38–44]. All rats successfully completed the early stages of pre-training in the reversal learning task where they had to learn to collect food from the magazine and to touch stimuli presented on the touchscreen to earn rewards, achieving these to criterion in a similar number of sessions (Magazine Training (mean ± s.e.m): WT = 2.7 ± 0.5, $Cyfip1^{+/-}$ = 3.7 ± 0.3, ANOVA: GENOTYPE ($F(1,15) = 2.79$, $p = 0.12$); Touch Training: WT = 15.6 ± 2.2, $Cyfip1^{+/-}$ = 13.3 ± 2.1, GENOTYPE

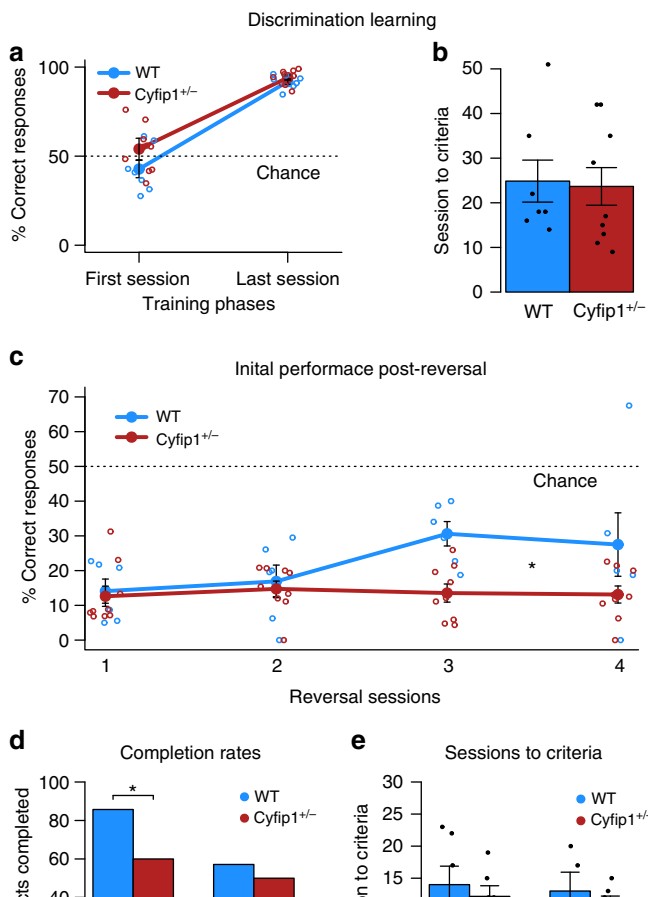

**Fig. 5** *Cyfip1* haploinsufficient rats show deficits in behavioural flexibility in a reversal learning paradigm. WT ($n = 7$) and *Cyfip1*$^{+/-}$ ($n = 9$) rats successfully acquire the visual discrimination in the touchscreen boxes, reaching the same level of performance (% correct) **a** during the last session of training, and **b** reaching criteria in a similar number of sessions (ANOVA: $F(1,14) = 0.03$, $p = 0.86$). One *Cyfip1*$^{+/-}$ rat did not complete initial learning. **c** *Cyfip1*$^{+/-}$ rats show more persistent responses to the original stimulus response contingencies during the first few sessions of reversal, (WT ($n = 6$) and *Cyfip1*$^{+/-}$ ($n = 9$), ANOVA: GENOTYPE X SESSION interaction, $F(3,39) = 3.76,*$), where one of the WT rats did not start reversal learning task. **d** A larger proportion of *Cyfip1*$^{+/-}$ rats failed to reach the 50% correct criterion during reversal than WT rats, (Chi-squared: $\chi^2 = 9.61,*$). **e** However, those that do reach criteria do so in a similar number of sessions as WT rats (effect of GENOTYPE, ANOVA: 50% criteria ($F(1,10) = 0.31$, $p = 0.59$), and 80% criteria ($F(1,7) = 0.44$, $p = 0.57$). Data are mean ± SEM. *<0.05, **<0.01, ***<0.001. The raw number of animals completing each task can be seen in Supplementary Table 1a. Source data are also provided as a Source Data file

($F(1,15) = 0.53$, $p = 0.48$)). Then the rats moved on to the visual discrimination training where the two stimuli were present. As illustrated in Fig. 5a, both groups achieved high levels of performance to a criterion of 80% correct trials across two successive days and there was no difference in the number of sessions required to reach criterion between the groups (Fig. 5b, ANOVA: GENOTYPE ($F(1,14) = 0.03$, $p = 0.86$)). One *Cyfip1*$^{+/-}$ was not included in these and subsequent analyses as they failed to

reach criterion on the visual discrimination task (see also Supplementary Table 1a).

Following acquisition of the initial visual discrimination the contingencies were reversed. This manipulation had the expected effect where perseverating with the previously correct choice led to scores were that were below chance levels (i.e., 50%). However, in contrast to the basic learning of the visual discrimination, there was an effect of genotype when the reward contingencies were reversed in the immediate post-reversal sessions (see Fig. 5c). In this part of the task,[45,46] the *Cyfip1*$^{+/-}$ rats continued to respond to the original S + over successive sessions, while the WT began to switch their choices to the new S + (a 2 × 2 ANOVA revealed a main effect of SESSION ($F(3,39) = 3.67$, $p < 0.05$) and a GENOTYPE × SESSION interaction ($F(3,39) = 3.76$, $p < 0.05$), where a main effect of GENOTYPE was marginally significant ($F(1,13) = 4.32$, $p = 0.06$)). Further analysis of simple main effects revealed an effect of SESSION in the WT ($F(3,39) = 7.34$, $p < 0.01$) but not in the *Cyfip1*$^{+/-}$ group ($F(3,39) = 0.09$, $p = 0.96$), and an effect of GENOTYPE on sessions 3 and 4 following reversal (Minimum $F(1,52) = 9.42$, $p < 0.01$).

In subsequent sessions with the reversed contingencies the rats gradually reached a criterion of 50% correct. However, fewer *Cyfip1*$^{+/-}$ than WT rats reached this criterion (Fig. 5d, Chi-squared: $\chi^2 = 9.61$, $p < 0.05$). However, again, group differences were apparent in those fewer *Cyfip1*$^{+/-}$ rats that successfully reached this criterion (Fig. 5d, Chi-squared: $\chi^2 = 9.61$, $p < 0.05$). Here, three *Cyfip1*$^{+/-}$ rats failed to complete early reversal compared to only one WT. Effectively this sub-group of rats were never able to successfully inhibit the previously learned response despite being given ample opportunity to do so that extended to the end of the experiment (rats that completed this phase of reversal did so in an average of 13 sessions, whereas those that failed to learn the reversed contingencies had an average of 22 sessions before the end of the experiment). Figure 5d also illustrates the high degree of behavioural specificity shown by the *Cyfip1*$^{+/-}$ rats in the task; insofar as genotype differences were not evident in the relative proportion of those rats that were able to successfully inhibit the previous response strategy, and went on to learn the new contingency to 80% criterion (two WT rats, and one *Cyfip1*$^{+/-}$ rat failed to reach 80% criterion and so were not included in subsequent analyses of sessions to criterion). Moreover, *Cyfip1*$^{+/-}$ rats that completed these stages of reversal (reaching 50 and 80% correct) did so in a similar number of sessions to WTs, where separate ANOVAs revealed no effect of GENOTYPE on sessions to either 50% criteria ($F(1,10) = 0.31$, $p = 0.59$), or 80% criteria ($F(1,7) = 0.44$, $p = 0.57$) (Fig. 5e; see also Supplementary Table 1a for the number of rats completing each stage of the reversal task; the total sessions and trials across the whole task are also shown in Table 1b).

We further assessed the effects of *Cyfip1* haploinsufficiency on the ability to change behaviour in the face of changed contingencies (WT $n = 21$, *Cyfip1*$^{+/-}$ $n = 15$) using an established associative mismatch task[47–50] when the contingencies involving sensory events were changed. Here, rats first received two audio–visual sequences (i.e. Tone→Steady Light, Click→Flashing Light; the combinations were counterbalanced across animals). As reported previously[47–50], presentations of the initially novel visual stimuli resulted in an orienting response towards the light cue that habituated over the course of the four training sessions, as animals came to expect the presentation of the light following the auditory cue. There were no effects of genotype on behaviour in this part of the task or on habituation to the test apparatus, Fig. 6a, where a 2 × 2 ANOVA on activity during habituation to the chambers revealed a main effects of BLOCK ($F(5,170) = 119.5$, $p < 0.05$) but no other effect of genotype or interaction (Maximum $F(5,170) = 0.4$, $p = 0.85$). A

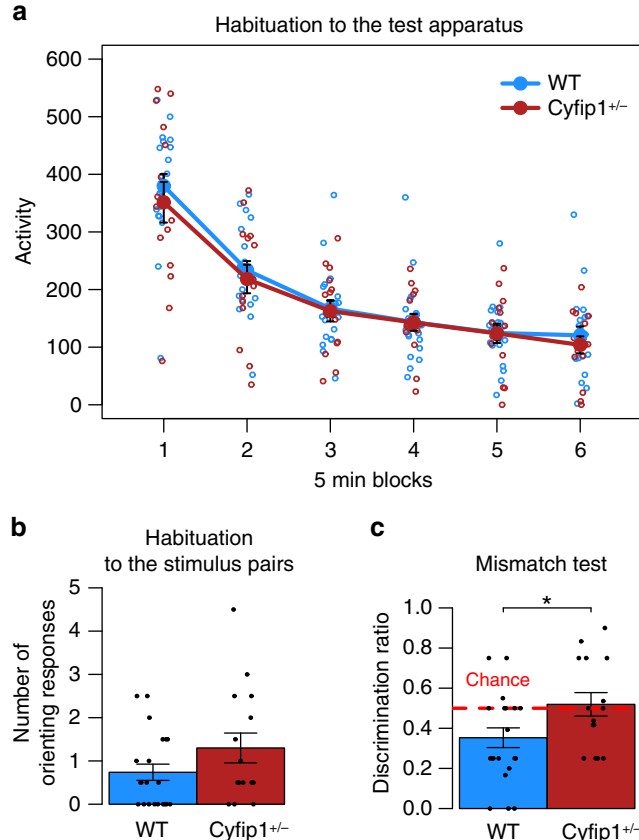

**Fig. 6** *Cyfip1* haploinsufficient rats show deficits in flexible responses in orienting behaviour on a mismatch task. **a** There are no genotype effects on habituation to the experimental apparatus. Both WT ($n = 21$) and *Cyfip1*[+/−] ($n = 15$) show reduced activity to the experimental chambers over the course of each 30-min long session (5 min blocks). **b** Both WT and *Cyfip1*[+/−] rats showed reduced orienting to the auditory-visual sequences presented during training (ANOVA: $F(1,34) = 2.4$, $p = 0.62$), and **c** WT rats preferentially responded to the novel mismatched visual stimuli over the familiar matched stimuli (discrimination ratio <0.50). In contrast, *Cyfip1*[+/−] rats showed no preference, responding equally to both matched and mismatched visual cues (ANOVA: $F(1,34) = 5.92$,*). *<0.05, **<0.01, ***<0.001. Source data are provided as a Source Data file

one-way ANOVA revealed no differences in orienting to the cued visual stimuli at the end of the training phase (Fig. 6b, GENOTYPE ($F(1,34) = 2.4$, $p = 0.62$)).

During testing rats were presented with both the original training trials (match; i.e. Tone→Steady Light, Click→Flashing Light) and novel mismatch combinations of the same cues (e.g. Tone→Flashing Light, Click→Steady Light), WT rats showed the normal preference[47–50] to orient to the visual cues on mismatch trails (relative to match trails) demonstrating their ability to respond to the changed contingencies. In contrast, the *Cyfip1*[+/−] rats failed to show this preference, instead showing no preference for responding to either the habituated (match) or the novel (mismatch) cues (Fig. 6c, ANOVA: GENOTYPE ($F(1,34) = 5.92$, $p < 0.05$) as indicated by the discrimination ratios (total orienting to matched/total orienting to both matched + mismatched). This pattern of behaviour represents further evidence of a deficit in behavioural flexibility associated with *Cyfip1* haploinsufficiency. In this case, the deficit was revealed by a failure to respond when the contingencies involving sensory events were changed.

## Discussion

We used a CRISPR/Cas9 engineered rat line to model the contribution of *Cyfip1* haploinsufficiency to white matter changes observed in carriers of the pathogenic 15q11.2 BP1-BP2 deletion. The *Cyfip1*[+/−] rat model allowed us to carry out a DTI analysis with high resolution using identical preprocessing to our 15q11.2 BP1-BP2 human imaging study, and employ rigorous statistics including exploratory voxel-wise assessments permitting comparisons of genotype effects across brain regions. The *Cyfip1*[+/−] rat line provided enhanced experimental tractability in terms of direct access to brain tissue, with interpretation of DTI changes at the cellular and molecular level, and also allowed relevant behavioural analyses under controlled conditions.

A main finding of the DTI experiments were significant decreases in FA that were most pronounced in the corpus callosum and external capsule. More widespread changes in white matter were apparent when using a less conservative correction method including increased FA in some areas of the fornix/fimbria. The precise relationship between DTI measures and cellular changes is subject to ongoing debate[24] and whilst DTI can identify white matter changes it cannot definitively distinguish between disruptions to axons and/or myelin. Consequently, at the outset the DTI effects we obtained in the rat model could have been related to changes in axon microstructure or myelin, or both.

Transmission electron microscopy indicated a thinning of the myelin sheath in the corpus callosum of the *Cyfip1*[+/−] rats in the absence of any changes in axonal number or diameter. Myelin is produced at the end stages of oligodendrocyte differentiation and we found reductions in later stage mature oligodendrocytes in corpus callosum in-vivo. However, whilst previous evidence in other contexts has shown that changes in oligodendrocyte number can be associated with myelin thickness[51,52] this still leaves the question of the cellular mechanism(s) by which *Cyfip1* haploinsufficiency could interfere with the process of myelination and lead to myelin thinning.

We addressed the issue of cellular mechanism taking advantage of the enhanced cellular resolution offered by cultured primary oligodendrocyte. We showed evidence that *Cyfip1* haploinsufficiency impacts on mature oligodendrocyte function by hindering the intracellular distribution of a key protein, MBP, where in the *Cyfip1* mutants the protein localised to the cell body and failed to achieve the normal highly distributed pattern of expression encompassing the distal cell processes and membranes. The myelination process is initiated by events occurring at the distal parts of the highly branched mature oligodendrocyte and hence, our finding of a deficit in translocation of MBP to those regions is consistent with and provides a mechanism for our in vivo observations of myelin thinning in the *Cyfip1*[+/−] animals. Several previous studies have established a critical role for MBP in initiating and supporting the production of myelin in mature oligodendrocytes triggered by contact with axons, where a principle function of MBP is to organise the compaction of oligodendrocyte plasma membranes to form myelin[53]. However and of key relevance, previous studies such as those reporting the effects of mutations Kif1b[29] have also demonstrated myelin deficits, including myelin thinning, that are associated with abnormal intracellular distributions of MBP highly reminiscent of the pattern seen in the *Cyfip1*[+/−] animals.

The effects of *Cyfip1* haploinsufficiency on oligodendrocytes and myelination may be related to deficits in actin physiology via interactions with the WAVE complex. Multiple aspects of oligodendrocyte function, including cell proliferation, differentiation, migration and myelination are reliant on effective cytoskeleton remodelling[10]. In mouse models manipulating

WAVE1 protein directly impacts on the ability to form myelin in the corpus callosum[12]. MBP stimulates the production of soluble G-actin from F-actin as part of the mechanism controlling the necessary disassembly of actin filaments in the distal processes of oligodendrocytes[11] which is a crucial step for myelination[54]. Consequently, our findings showing a failure of MBP to be located appropriately precisely where these interactions take place would impact adversely on myelin formation. Whether any effects mediated by MBP abnormalities seen here also involve the previously reported aberrant F-actin dynamics[55] in Cyfip1[+/−] mouse models, remains to be established.

Effects related to the close interaction between CYFIP1 and FMRP affecting protein translation cannot be discounted, especially in view of a degree of overlap between white matter changes in the Cyfip1[+/−] rat model and a mouse Fmr1 knockout, specifically reduced FA in the corpus callosum[56]. The mouse Fmr1 knockout also revealed evidence of global disruptions in functional connectivity as indexed by fMRI[56]. Prior to myelination, mRNAs encoding MBP are transported into the oligodendrocyte processes, where local translation of MBP mRNAs occurs, so perturbations in CYFIP1-FMRP interactions could presumably affect MBP mRNA translation. However, whilst there is some in vitro evidence linking FMRP with regulation of mRNA translation in developing oligodendrocytes[57], current data does not allow further speculation regarding FMRP mediated effects on the phenotypes seen in the Cyfip1[+/−] rat model. Additional complexity is apparent when considering the range of functions that may be sensitive to reduced dosage of Cyfip1, encompassing oligodendrocyte production, differentiation and migration; effects on neurons as opposed to oligodendrocytes, can also have an impact on axon-oligodendrocyte interactions and influence myelination in that way[58].

We assessed the extent to which the imaging and cellular data were associated with behavioural effects. Behavioural flexibility has been shown to be sensitive to abnormalities in myelination in mouse models of myelin deficits[15]. Furthermore, in humans, corpus callosum morphology correlates with behavioural flexibility in a study using a cohort of twin pairs, where one was diagnosed with bipolar disorder and the other was clinically healthy[30]. We therefore focused the behavioural analyses on psychological processes supporting such flexibility. We first showed that during reversal learning, Cyfip1[+/−] rats showed greater early perseveration in choosing the previously rewarded stimulus. The effects in the Cyfip1[+/−] rats were highly specific to the reversal element of the task, with no concomitant effects on initial learning.

The pattern of normal initial learning but impaired behavioural flexibility was also evident when assessed independently in an associative mismatch task. In this case, the changed contingencies involved the relationships between the components of audio–visual sequences. This task has been described previously[47–50], and characterised with respect to its psychological origin[47,49], the conditions under which it is observed[48,50] and its underlying brain substrates, where there is evidence that hippocampal circuitry is involved, and mediation of related psychological processes by callosal circuitry[59–61]. We again observed a highly specific effect of Cyfip1 haploinsufficiency on the orienting behaviour generated when there was a mismatch between the trained and tested audio–visual sequences, with the Cyfip1[+/−] rats not showing the expected increase in orienting to a familiar visual stimulus that was unpredicted by the auditory cue that preceded it. An inability to alter behaviour in response to changes in the environment has been strongly associated with orbitofrontal cortex (particularly during reversal learning[62]) and ventral prefrontal cortex damage in humans[63] and rats[62,64]. Whilst the role of the corpus callosum in behavioural flexibility has not been

as extensively investigated, it is known to be important in the functional integrity of brain regions that are classically linked with behavioural flexibility[32,64,65]. The corpus callosum (along with the internal capsule) carries white matter bundles containing axons projecting from the frontal cortex and striatal regions[32], and pruning and myelination of the corpus callosum coincides with cortical maturation in the frontal cortex, mutually influencing each other's development[65,66].

Increases in FA with little evidence of reductions were a prominent feature of our findings in human 15q11.2 BP1-BP2 deletion carriers[7]. Hence, while the rat model and human phenotype converged on white matter changes they differed in the direction of the change. Differences between the human and rat findings could have resulted from several factors. First, the 15q11.2 BP1-BP2 deletion involves three other genes in addition to CYFIP1, especially NIPA1 which is expressed in the brain and was found to inhibit the bone morphogenic protein (BMP), crucial for typical axonal growth and guidance[67,68]. Therefore, a priori, haploinsufficiency of NIPA1, and possibly the other genes in the interval[69,70], may impact on the 15q11.2 BP1-BP2 deletion DTI phenotype. The possibility that there are species differences in the expression patterns of CYFIP1/Cyfip1 and also any compensatory responses to haploinsufficiency should also be borne in mind. Furthermore, the humans and rats are likely to have been subject to differential compensatory mechanisms arising from very different environmental challenges across their lifespan[6,71,72]. Moreover, as changes in myelin thickness impact relatively modestly on DTI measures[24] it may be that whilst myelin changes may be present in both human and the rat model, in terms of the human DTI data any effect on myelin may have been masked by other molecular and cellular consequences of the copy number deletion. To date, there have been no published studies of myelin (as opposed to overall white matter) changes in 15q11.2 BP1-BP2 deletion, though the current data predicts their existence and this is something that could be tested using ultrastructural magnetic resonance imaging (MRI) methods providing the necessary resolution to visualise and quantify myelinated axons directly in the living human brain[73]. Nonetheless, whilst an exact between-species comparison would require an assessment of CYFIP1-specific heterozygous humans, we have demonstrated a clear link between Cyfip1 haploinsufficiency and white matter microstructure.

In conclusion, we have employed a novel rat model of Cyfip1 haploinsufficiency to probe the neurobiological and behavioural mechanisms underlying the significantly enhanced risk for psychopathology linked to the 15q11.2 BP1-BP2 deletion. We found disturbances to white matter as seen in human carriers, and showed effects on myelin thickness associated with an abnormal intracellular distribution of MBP, together with evidence of highly specific effects on behavioural flexibility. Cross species comparison of the imaging phenotypes in rats and humans suggest it is unlikely that effects mediated by CYFIP1 are solely responsible for the 15q11.2 phenotype, and additional work is required to determine the contribution made by the other three genes, NIPA1, NIPA2, and TUBGCP5 affected in the 15q11.2 BP1-BP2 deletion. However, these findings in the Cyfip1 rat model give an insight into the contribution made by low dosage of CYFIP1 to the 15q11.2 BP1-BP2 deletion phenotype.

## Methods

**Rats**. The Cyfip1 rat model was created by Cardiff University in collaboration with Horizon Discovery (St Louis, USA) using CRISPR-Cas9 targeting (https://www. horizondiscovery.com/) and supported by a Wellcome Trust Strategic Award (DEFINE). Full information on the creation and validation of the rat model is in the Supplementary Methods section. All the rats used in this study were Long Evans males. The rats were produced from breeding stocks held at Charles River (UK) using a WT x HET design resulting in an average 1:1 WT to Cyfip1[+/−], the

mutation was transmitted in Mendelian fashion with no sex bias and the rats were healthy and viable showing no general ill effects of the mutation. The rats were transported to Cardiff at 8–10 weeks of age. At Cardiff the rats were housed in mixed-genotype groups of 2–3 rats. The rats had free access to food and water (except for those used in the reversal learning task, see below) and lived under the condition of a 12 hr light/day cycle (lights on at 7:00 am), room temperature $21 \pm 2\,°C$. Rats used in DTI were 5-months-old. The rats were euthanised 1 month after the scanning and used for immunofluorescence. The rats used for electron microscopy were 6-months-old. The rats used for behavioural experiments were 6–9-months-old. The reversal learning task was motivated by liquid reward (10% sucrose solution w/v) and to enhance working in the task, the rats were subject to water restriction immediately prior to and during task training, in which case the rats were given 2 h access to water per day. The water restriction schedule has no adverse effects on the health or welfare of the rats, being designed to give rise to a temporary increase in motivation for the liquid reinforcement, and across the whole day the rats on the schedule drink as much fluid as under free access conditions. All the experimental procedures were performed in accordance with institutional animal welfare, ethical and ARRIVE guidelines and under the UK Home Office License PPL 30/3135 (Animals (Scientific Procedures) Act 1986).

**Diffusion tensor imaging acquisition**. A cohort of 24 rats (WT $n = 12$ and Cyfip1+/− $n = 12$) were anaesthetised with isoflurane in oxygen at 4% and maintained at 1% during the scanning. MRI scans were acquired with a 9.4 T MRI scanner (Bruker, Karlsruhe, Germany) with a 30-cm bore and a gradient strength of up to 600 mTm$^{-1}$. The MRI protocol included DTI acquisition with a diffusion-weighted (DW) spin-echo echo-planar-imaging (EPI) pulse sequence having the following parameters: TR/TE = 4000/22 ms, $\Delta/\delta$ = 10.5/4.5 ms, two EPI segments, and 60 noncollinear gradient directions with a single b-value shell at 1000 smm$^{-2}$ and one image with a b-value of 0 smm$^{-2}$ (referred to as b0). Geometrical parameters were: 34 slices, each 0.32 mm thick (brain volume) and with in-plane resolution of $0.32 \times 0.32$ mm2 (matrix size 80 × 96; FOV 25.6 × 30.73 mm$^2$). The DTI protocol lasted ~16 min. In addition, high resolution, T2 weighted images were acquired for anatomical reference with a multi-slice multi-echo pulse sequence with the following parameters: TR of 7200 ms, TE of 15 ms and effective TE of 45 ms, rare factor was 8. Image resolution was set to 0.22 mm$^3$ with matrix size of 128 × 160 × 50 to cover the entire brain.

**DTI data correction and DTI maps extraction**. ExploreDTI 4.8.3[74] and SPM (version 12, UCL, London, UK) were used in the preprocessing of the rat DTI data. First, eddy-current induced distortion and motion correction were performed and mean-DWI images were extracted using ExploreDTI. Non-brain tissue was removed from the mean-DWI and the $T_2$-weighted images following these steps: (1) $T_2$-weighted scans were anisotropic smoothed using ExploreDTI, (2) both smoothed $T_2$-weighted and mean DWI images were bias corrected using the segmentation tool in SPM12, (3) the bias corrected $T_2$-weighted were coregistered with a population-specific template and multiplied by a mask to remove the non-brain tissue, (4) the skull was removed from the mean DWIs using the 3D masking option in ExploreDTI. Then, data was corrected for field inhomogeneities, using ExploreDTI, where the skull-stripped mean DWIs images were used as a native space mask, and the skull-stripped $T_2$-weighted structural scans were used as transformed space mask. Each DWI image was nonlinearly warped to the $T_2$-weighted image using non-DWIs map as a reference. ExploreDTI was used to generate FA, AD, RD and MD maps.

**Preprocessing for Tract-Based Spatial Statistics**. For the voxel-wise analyses of DTI data, Tract-Based Spatial Statistics (TBSS) method was implemented, which is part of the FSL. All FA maps were submitted to a free-search for a best registration target, where each volume was first registered to every other volume, and the one requiring minimum transformation to be registered to other volumes was selected as the best registration target. This target was used as a template into which the registration was performed. Following registration, a mean FA map was calculated, thinned to represent a mean FA skeleton, and an optimal threshold of 0.2 was applied to the mean FA skeleton to create a binary white matter skeleton mask (Supplementary Fig. 1). The local FA-maxima, as well as the AD, RD and MD, of each rat were projected onto this white matter skeleton.

**ex vivo transmission electron microscopy and immunofluorescence**. For transmission electron microscopy, a new cohort of nine rats (WT $n = 5$ and Cyfip1+/− $n = 4$) was used. For immunofluorescence seven brains were randomly selected (WT $n = 7$ and Cyfip1+/− $n = 7$) from the cohort used for DTI. In both cohorts, the rats were intracardially perfused with 0.1 M phosphate buffered saline (PBS), followed by 4% of glutaraldehyde in 0.1 M PBS in the cohort used for electron microscopy, and 4% paraformaldehyde in 0.1 M PBS (PFA) in the cohort used for immunofluorescence. For transmission electron microscopy, the brains were placed on a shaker to postfix in glutaraldehyde for 4 h, after which they were placed in phosphate buffered saline and stored at 4 °C until further use. Then the brains were embedded in TAAB embedding resin. Ultra-thin sections (50 nm) were stained with aqueous 4% uranyl acetate and lead citrate. The sections were visualised on a transmission electron microscope (CM12, Philips, the Netherlands)

and, for quantification, images were taken using an on-axis 2048 × 2048 charge-coupled device camera (Proscan, Schering, Germany). In order to obtain a representative sample, 15 regions across the extent of the anterior-posterior extent of the corpus callosum per animal were taken for quantification. For immunofluorescence, the brains were placed on a shaker to postfix in PFA for 4 h, after which they were placed in phosphate buffered 30% sucrose. Coronal cryosections of the brain, of 15 μm thickness, were made on a cryostat (CM1860 UV, Leica, UK), mounted onto a Poly-L-Lysine (PLL)-coated slides (three sections per slide), and stored at −20 °C. For immunofluorescence, antibodies were used as follows: anti-Olig2 (ab109186, Abcam) 1:400, anti-APC [CC-1] (ab16794, Abcam) 1:400, anti-MBP (MAB386, Millipore) 1:300. For the Olig2 and Cc1 doublestaining, the slices were heated in a 5% citrate buffered antigen retrieval solution (pH 6, 10x, Sigma-Aldrich Company, UK), using a water bath at 90% for 10 min. All the slices were blocked for 1 h with 5% donkey serum (Sigma-Aldrich Company, UK), and 0.3% Triton X-100 in PBS. The appropriate primary antibodies were applied and incubated overnight at 4 °C. On the next day, after washing, the slices were incubated for 2 h with secondary antibodies (Alexa Fluor Life Technologies, Manchester, UK), in a concentration of 1:1000 at room temperature. Then, the slides were washed, counterstained with 1:1000 DAPI, mounted and cover-slipped. For quantification of Olig2 + and Cc1 + cells, images were taken on an inverted fluorescent time lapse microscope (DMI6000B, Leica, UK), and at least 4 images from random visual fields were taken from regions including the corpus callosum and external capsule. For quantification of MBP intensity, one coronal section per rat was taken on an Axio scan (Zeiss, Germany), and the same exposure time and intensity were used for all the slides.

**in vitro immunofluorescence**. Primary OPC cultures were isolated from neonatal Long Evans (postnatal day 0–3) rat from cortices following a standard protocol[26]. This protocol is known to generate OPC at ≥95%. Briefly, cerebral cortices were dissected, and the meninges were removed. Following the enzymatic digestion for an hour, the cell suspension was placed into cell culture flasks. The mixed glial cultures were grown for ~10 days in Dulbecco modified Eagle medium supplemented with 10% foetal calf serum at 37 °C in 7.5% CO$_2$. On day 10, the flasks were shaken for 1 h at 260 rpm on an orbital shaker to remove the loosely attached microglia and were then shaken at 260 rpm overnight to dislodge the loosely attached oligodendrocyte precursors. OPCs were seeded onto PLL-coated eight well chamber slides ($2 \times 10^4$ cells/well), in Sato's medium supplemented with 0.5% foetal calf serum (FCS) in order to induce differentiation. After 3 days of differentiation, the cells were fixed using 4% PFA in PBS for 10 min following two washes with PBS. Cells were stained with anti-O4 (1:200; MAB345, Millipore) and anti-MBP (1:200; MAB386, Millipore) antibodies (Alexa 555/488-conjugated secondary antibody, 1:300; Alexa Fluor Life Technologies). For quantification, images were taken on an inverted fluorescent time lapse microscope (DMI6000B, Leica, UK) with ×20 magnification, where 5 images from random visual fields were taken per well. Three independent biological replicates were performed.

**Reversal learning**. A separate group of 17 rats were used in the reversal learning task (WT $n = 7$, Cyfip1+/− $n = 10$). Testing was conducted in a touchscreen-based automated operant system that consisted of an operant chamber with a flat-screen monitor equipped with an infrared touchscreen with accompanying Animal Behaviour Environment Test (ABET) II software (Campden Instruments, Leics). Session duration was 30 min, or until 100 trials were completed under all training conditions. Pre-training consisted of two stages (Magazine Training and Touch Training) these gradually shaped the screen-touching behaviour required for the reversal learning touchscreen task proper. Following successful completion of pre-training Visual Discrimination Training began (Supplementary Fig. 4); two stimuli were presented at a time (S+ and S−, counterbalanced across animals), on either side of the screen. The rat had to touch the correct stimulus (S+) to elicit reward. Reward delivery was accompanied by illumination of the tray light and a tone. Entry to collect the reward turned off the tray light and started the inter-trial interval (ITI −5s) following which the rat initiated the next trial by a second magazine entry. Touching the incorrect stimulus (S−) terminated the trial and the house-light was turned on for a time-out period of 5 s and no reward given, following the time-out the ITI period began after which the rat had to initiate the next trial by executing a magazine entry. Once the rats reached performance criteria (Completing 50 + trials with 80–85% correct, for 2 consecutive sessions), the contingencies were reversed (previous S+ now S−; previous S− now S+) and behaviour monitored (see Supplementary Figure 4 for schematic of task design).

**Mismatch task**. A different group of 36 rats performed the mismatch task (WT $n = 21$, Cyfip1+/− $n = 15$). On the first 4 days, the rats were placed in the experimental apparatus (a modified skinner box allowing presentations of auditory and visual stimuli) for 30 min. Following this general habituation to the apparatus they received 4 days of training with two audiovisual sequences. One auditory stimulus (a 2 kHz tone) preceded the constant presentation of a light, whereas a second auditory stimulus (a 10 Hz series of clicks) preceded the flashing presentation of the same light stimuli (i.e. Tone→Steady Light, Click→Flashing Light; the combinations were randomly counterbalanced across animals). All stimuli were presented for 10 sec. There were 10 presentations of both audiovisual sequences on each of the first

3 days of training and six presentations of both sequences on day 4 that served as warm-up trials for the eight test trials that immediately followed. The inter-trial-interval was 2 min. Rats received two types of test trials, match and mismatch. The order in which the two types of test trials were presented was counterbalanced. Match test trials were presentations of the same audiovisual sequences that had been presented during training (e.g. Tone→Steady Light, Click→Flashing Light), whereas on mismatch trials the auditory stimuli preceding the visual stimuli were exchanged (e.g. Tone→Flashing Light, Click→Steady Light). All experimental sessions were recorded using a video recorder and orienting responses subsequently scored by observers who were blind to the genotype of the rats and the nature of the test trials (match or mismatch). An Orienting Response (OR) was defined as the tip of a rat's snout being located in the side of the apparatus that contained the light and pointing in the direction of the light.

**Statistical analyses**. Differences in DTI measures between the two groups (WT and $Cyfip1^{+/}$) were assessed using voxel-wise independent t-tests, where two different contrasts were used (WT > $Cyfip1^{+/-}$, and $Cyfip1^{+/-}$ > WT). Using the randomise function (part of FSL), the null distribution was built over 1000 random permutations, using the TFCE[18] algorithm where cluster-like structures are enhanced, and the results are shown for $p < 0.05$. For multiple comparison correction, first FWE correction was used. Since only FA changes were found within this analysis, we also used a less conservative correction method based on FDR correction, purposed by Benjamini–Hochberg[21]. To quantify the changes in areas where significant differences in FA were seen after FWE correction, regions of interest (ROIs) were manually delineated using FSL. Several consecutive slices were outlined on the coronal plane, and the selected ROIs included the corpus callosum, internal capsule, external capsule, and fornix/fimbria regions. The CBJ13 MR-histology rat atlas at age P80[75] was used as reference. A representation of the binary masks can be found in Supplementary Fig. 2. FA, AD, RD and MD were quantified by applying these binary masks and extracting the mean values for each region across subjects.

For quantification of cells the ImageJ software (version 1.51) was used. The number of myelinated and unmyelinated axons, axon diameter, myelin thickness and g-ratio (measure of myelin thickness relative to axon diameter: where lower g-ratios indicates thicker myelin sheath) of normally myelinated axons were quantified. A total of 13127 (WT $n = 7148$, $Cyfip1^{+/-}$ $n = 5979$ axons) myelinated axons were analysed. For quantification of oligodendrocytes ex vivo, the total number of Olig2 + , and the overlapped Olig2 +/Cc1 + cells were counted. Only cell bodies clearly identified by Olig2 and Cc1 immunofluorescence and overlapping with DAPI staining were counted. MBP + reactivity was determined by comparing immunofluorescence staining intensity. The whole region of corpus callosum and external capsule was selected in the coronal section, and quantification was done by calculating the mean intensity of the pixels above a preset intensity threshold, multiplied by the number of pixels above that threshold, and dived by the total area quantified. For in vitro quantification, to assess OPC differentiation, the percentage of O4 + and MBP + cells relative to Hoechst-stained nuclei were quantified. In order to compare different levels of maturation of oligodendrocytes, MBP-positive oligodendrocytes were classified into three categories considering the distribution of MBP in the cells: (i) type 1, where the MBP staining was only present in the nucleus, (ii) type 2, ramified distribution and (iii) type 3, membranous distribution. Furthermore, to quantify MBP distribution in type 2 and type 3 cells, the area of MBP + staining was quantified.

All the cell quantifications were conducted with the investigator blinded to the phenotype. Differences between WT and $Cyfip1^{+/-}$ were analysed in RStudio version 1.1.463 (R Foundation for Statistical Computing, Vienna, Austria). In order to compare all the axons in each group while taking into account variation across individuals, we used linear mixed effects models to analyse the effect of genotype on axon diameter, g-ratio and myelin thickness, where these measures were considered fixed effects, and animals were considered random effects. Since we only had one random effect, we used non-restricted maximum likelihood to estimate the model parameters. In this analysis, the myelin thickness was log-transformed since the data followed a log-normal distribution, whereas the other measures followed a normal distribution. For in-vitro analyses, linear mixed effects models were also used to consider variation across biological repeats (where these were considered random effects). All the other measures were analysed using two-tailed unpaired Student's t-test. Data are given as mean ± s.e.m.

Visual discrimination and reversal learning performance was assessed using ANOVA with factors of GENOTYPE and SESSION. Any significant interaction was subsequently examined by analysing the Simple Effects. Completion rates for the rats during the different phases of reversal were assessed non-parametrically using Chi-squared test. Performance in the mismatch task was assessed using ANOVA with factor of GENOTYPE and BLOCK during the habituation to the test apparatus phase of training and factor GENOTYPE in the habituation to the stimulus pairs and mismatch test phases. Orienting responses in the mismatch test phase were analysed as a discrimination ratio (total orienting to matched/total orienting to both matched + mismatched).

**Reporting summary**. Further information on research design is available in the Nature Research Reporting Summary linked to this article.

## Data availability
All data from this study are available from the corresponding author upon reasonable request. The source data underlying Table 1, Fig. 2c–g, Fig. 3a–b, Fig. 4c–d, Fig. 5a–e, Fig. 6a–c, and Supplementary Figs. VI and VII are provided as a Source Data file.

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

## Acknowledgements

This work was also supported by a Wellcome Trust Strategic Award 'DEFINE' grant no. 100202/Z/12/Z and core support from the Neuroscience and Mental Health Research Institute, Cardiff University, UK. We acknowledge technical support for the DTI studies from Andrew Stewart, School of Biosciences, Cardiff University and excellent animal husbandry and care by the Joint Biological Services Unit personal at Cathays Animal

Facility, Cardiff University. We also acknowledge technical support for the transmission electron microscopy acquisition from Dr. Christopher Von Ruhland, Central Bio-technology Services, Cardiff University.

## Author contributions

A.I.S., J.E.H., Y.A.S., D.E.J.L., J.H. and L.S.W. planned, designed and instigated the study. S.T. conducted the molecular specification of the novel heterozygous *Cyfip1* rat line. Y.P., Y.A. and J.C. conducted the DTI imaging acquisition, and A.I.S., Y.A., D.E.J.L., J.H. and L.S.W. analysed the DTI data. A.I.S., Y.A.S. and N.H. obtained and analysed the transmission electron microscopy data, immunofluorescence, and cell culture data. J.E.H., A.I.S., J.H., T.E.L., R.C.H., T.H. and L.S.W. carried out and analysed the behavioural studies. A.I.S., J.E.H., J.H., M.J.O, D.E.J.L., R.C.H. and L.S.W. wrote and reviewed the paper.

## Additional information

**Competing interests:** The authors declare no competing interests.

