## [Transparent Peer Review File · Nature Communications]

Reviewers' comments:

Reviewer #1 (Remarks to the Author):

In the present study, Silva et al. investigated whether the pathological consequences observed in patients with a 15q11.2 copy number deletion may be explained by haploinsufficiency of one of the genes in this region, i.e. the risk gene CYFIP1. To this end, the authors generated a novel rat model of Cyfip1 haploinsufficiency using CRISPR/Cas9. They showed using diffusion tensor imaging and electron microscopy that myelination was reduced in the corpus callosum of these rats, and that this may be due to a reduction in mature oligodendrocytes that produce myelin. Moreover Cyfip1 haploinsufficiency resulted in an impairment in cognitive flexibility, one of the core behavioral deficits in individuals with schizophrenia and autism. Altogether, these data indicate that Cyfip1 haploinsufficiency may play an important role in the structural and behavioral consequences of the pathogenic 15q11.2 copy number deletion.

Overall this is a very solid study that addresses an important question, and the reported findings will be highly interesting to researchers in molecular psychiatry. The manuscript is very well prepared and includes an excellent introduction and a detailed and thoughtful discussion. However, the study would greatly benefit from additional experiments to address some of the mechanistic aspects of the consequences of Cyfip1 deletion as described below:

1. The authors postulate that the reduction in myelin thickness results from the reduction in the number of oligodendrocytes, and that this reduction in turn may be due to altered F-actin dynamics. Can the authors explain why this change in F-actin dynamics, which presumably occurs in every cell, has such a prominent effect specifically on oligodendrocytes and myelin formation, rather than causing more global changes? Is there any evidence that oligodendrocytes are more sensitive to CYFIP1-related changes in F-actin than neurons or other cells? Additional studies e.g. in cell culture would help to address this question.

2. What are the functional consequences of the reduced myelination for neuronal function and brain physiology? Is synaptic transmission changed?

3. The authors report a reduction in myelination as well as an impairment in cognitive flexibility, but they do not provide any experimental evidence that these two phenotypes are causally linked. Other changes in the Cyfip1 +/- animals could be causing the cognitive deficits. An interesting experiment to establish this link would be to reverse the myelin deficits, globally or specifically in the prefrontal cortex, and determine whether this rescues the behavioral impairments.

4. In the reversal learning task in Figure 4d, how do the authors explain that some Cyfip1 +/- rats were completely unable to learn the reversal, but those that did performed like normal WT rats? Is it possible that there are additional effects involved, e.g. off-target effects of the CRISPR strategy (despite the described testing for 10 potential OT sites)? Does the performance in the reversal learning task correlate with the reduction in myelination, possibly indicating a causal link?

Minor comment:

Figure 2a: Are the representative images for WT and Cyfip1 +/- on the same scale? The axons appear much larger in diameter in the WT than in the Cyfip1 +/- in this example, yet according to the quantification in Figure 2d there is no difference.

Reviewer #2 (Remarks to the Author):

Silva et al.

This paper is extremely well written and presented. The work is thorough and well thought out. The authors present a novel heterozygous *Cyfp1* rat model. They examined the rat with DTI to look for white matter microstructure differences as well as electron microscopy to investigate their DTI results and immunohistochemistry to further characterize the differences. I appreciate the extra care that was put into this study in order to confirm the FA findings with EM. I furthermore appreciate the statistical examination that was performed. My comments are relatively minor as I found the paper to be quite good as is.

- 1) The last sentence of the abstract made me expect some resting-state fMRI looking at functional connectivity or circuitry the way it is worded. Perhaps some mention of the behavioural tasks.
- 2) A b-value shell of 1000 s/mm² seems a little small for the rat. Why was this chosen?
- 3) Did you test for any behavioural/brain correlations in the touch screen task and the FA measurements? Or were they different cohorts?
 - a. I'm curious if you can see any notable FA differences across the *Cyfp1* (or WT for that matter) based on their performance in the tasks.
- 4) Did you do any correlations between the immunohisto and the FA measurements since they were on the same mice?

Reviewer #3 (Remarks to the Author):

Silva and colleagues have generated a novel innovative genetic rat model with *cyfp1* deficiency, an important protein for actin dynamics with links to schizophrenia and autism spectrum disorder. They used this novel rat model to assess a clinically relevant level of low dose of *Cyfp1* over other mouse models that KO. They present data on alterations in white matter microstructure by DTI, pronounced in the corpus callosum and external capsule. They also performed finer microscopy on these outcomes. Finally, they used advanced touchscreen cognitive behavioral testing to illustrate cognitive dysfunction on specific executive learning tasks.

Behavioral Critique:

There were a few major behavioral critiques.

It was disappointing that the same animals were not used for behavior and imaging, especially since the authors performed in vivo imaging. This was an obvious missed opportunity. It would seem a subset of the behavioral animals could have been tested for within subject correlations. In differing animals, how do we know the "level of circuitry impairment" would yield "level of behavioral impairment"

The nomenclature is off. The pairwise reversal learning task is just that "reversal learning", it is not go/no-go nor inhibition.

This text is incorrect and misleading "Successful reversal is demanding on attention and response inhibition, and can be viewed as a type of "go"– "no". The field would not agree on this statement. Reversal in the touchscreen or an operant box is mediated by the orbitofrontal cortex and OFC mediated reversal is specifically NOT response inhibition.

Schoenbaum G, Nugent SL, Saddoris MP, Setlow B (2002)

Orbitofrontal lesions in rats impair reversal but not acquisition of go, no-go odor discriminations. *NeuroReport* 13:885–890.

Schoenbaum G, Roesch MR, Stalnaker TA, Takahashi YK (2009b) A new perspective on the role of the orbitofrontal cortex in adaptive behaviour. *Nat Rev Neurosci* 10:885–892.

The neural basis of reversal learning: an updated perspective

A Izquierdo, JL Brigman, AK Radke, PH Rudebeck, A Holmes
Neuroscience 345, 12-26

Figure 4C should be illustrated as the field standard of early, mid, late reversal, as shown in Marquardt et al., 2017, 2018 and Brigman et al., Nature Neuro 2013.

Figure 4D shows that only 50% of WT completed the reversal to 80%, this is highly unusual especially for LE background strain. In our experience and prior literature, all animals plus or minus 1 complete, however it could be that they did not continue testing to go out to 30 days or more. This makes me think 4D may be an artifact of the shortened testing period.

The second behavioral assay is not well described by literature. Could the authors provide more referencing? If this is the first description, they should report how they validated this assay as measuring the complex processes that underlie it (lesion or pharm or dreadd blockage of signaling).

Why is the N so substantially larger for the second task?

Where are the control parameters that Lafayette provides you for motor? Trial number? Latency to collect reward (which tells us about motivation)

What do the authors think about that the circuitry that they've disrupted is not classically associated with these behaviors? Discussion should be added to expand their opinion of this and acknowledge the breadth of literature on the key neural correlate of the OFC in the reversal task.

Reviewer #1

We thank this reviewer for their comments and insights. We provide comprehensive replies to the points raised. In particular we have responded to the challenge of providing more mechanistic detail underlying the effects seen in myelin/oligodendrocytes by conducting, as suggested, a series of studies in cell culture. As detailed in point 1 below, we focused on the most immediate question of how *Cyfp1* haploinsufficiency impacts on the myelin abnormalities we observe. Our new data reveals a mechanism involving aberrant organisation of myelin basic protein in oligodendrocytes that is consistent with and provides an explanation for the effects on myelin thinning.

In the present study, Silva et al. investigated whether the pathological consequences observed in patients with a 15q11.2 copy number deletion may be explained by haploinsufficiency of one of the genes in this region, i.e. the risk gene CYFIP1. To this end, the authors generated a novel rat model of *Cyfp1* haploinsufficiency using CRISPR/Cas9. They showed using diffusion tensor imaging and electron microscopy that myelination was reduced in the corpus callosum of these rats, and that this may be due to a reduction in mature oligodendrocytes that produce myelin. Moreover *Cyfp1* haploinsufficiency resulted in an impairment in cognitive flexibility, one of the core behavioral deficits in individuals with schizophrenia and autism. Altogether, these data indicate that *Cyfp1* haploinsufficiency may play an important role in the structural and behavioral consequences of the pathogenic 15q11.2 copy number deletion.

Overall this is a very solid study that addresses an important question, and the reported findings will be highly interesting to researchers in molecular psychiatry. The manuscript is very well prepared and includes an excellent introduction and a detailed and thoughtful discussion. However, the study would greatly benefit from additional experiments to address some of the mechanistic aspects of the consequences of *Cyfp1* deletion as described below:

1. The authors postulate that the reduction in myelin thickness results from the reduction in the number of oligodendrocytes, and that this reduction in turn may be due to altered F-actin dynamics. Can the authors explain why this change in F-actin dynamics, which presumably occurs in every cell, has such a prominent effect specifically on oligodendrocytes and myelin formation, rather than causing more global changes? Is there any evidence that oligodendrocytes are more sensitive to CYFIP1-related changes in F-actin than neurons or other cells? Additional studies e.g. in cell culture would help to address this question.

We have conducted additional studies in cell culture to provide further evidence and mechanistic detail to the key question of how *Cyfp1* haploinsufficiency can impact on oligodendrocytes and myelination. We now have additional data, detailed in Figure 4, revealing that low dose *Cyfp1* impacts on mature oligodendrocytes, the myelin producing cells of the CNS, by hindering the internal distribution of a key protein, myelin basic protein (MBP), which is a marker for oligodendrocyte differentiation and essential for the production of myelin. The myelination process is initiated by events occurring at the distal parts of the highly branched mature oligodendrocyte and our finding of a constrained distribution of MBP, where in the *Cyfp1* mutants the protein

localises more to the cell body and fails to achieve the normal highly distributed pattern of expression encompassing the distal cell processes, is consistent with and provides a mechanism for the consequent effects on myelin thinning. Furthermore, there is much previous evidence linking actin dynamics to myelination and the key role played by MBP, supporting our speculation of there being a role for *Cyfp1* in this process. With respect to the more general point about the extent to which *Cyfp1* haploinsufficiency may impact on other cellular processes, we agree that it is unlikely that oligodendrocytes are uniquely affected, but our focus on myelin and oligodendrocytes is entirely justified on the basis of the imaging data showing changes in white matter microstructure. That is not to say that the comments made by the reviewer are invalid, they are valid in a general sense, but to have looked first at other cell types and biological processes would have been illogical on the basis of the phenotypes we were investigating. We prioritized our lines of enquiry to those most immediately relevant to explaining the changes in white matter microstructure revealed by the DTI analysis.

2. What are the functional consequences of the reduced myelination for neuronal function and brain physiology? Is synaptic transmission changed?

As we note in the main text, effective connectivity both within local circuits and between more distally located brain regions is crucial for the co-ordination of brain function at the synaptic level. Myelination plays an important role primarily by impacting on the speed of neural conduction which in turn is a key determinant of coherence between activity of neurons and the balance between excitatory and inhibitory input that regulates the firing probability of a neuron (Feldman, 2012; Markram et al., 2012). As well as normal functioning it is apparent that abnormalities in myelination, and consequent effects on connectivity, is an important feature of many CNS diseases, such as leukodystrophies, schizophrenia, multiple sclerosis, and amyotrophic lateral sclerosis, among an increasing number of others (Compston and Coles, 2008; Fields, 2008; Huang et al., 2015; Lee et al., 2012; Mighdoll et al., 2015; Miyata et al., 2015; Olmos-Serrano et al., 2016; Philips and Rothstein, 2014; Pouwels et al., 2014).

3. The authors report a reduction in myelination as well as an impairment in cognitive flexibility, but they do not provide any experimental evidence that these two phenotypes are causally linked. Other changes in the *Cyfp1*^{+/-} animals could be causing the cognitive deficits. An interesting experiment to establish this link would be to reverse the myelin deficits, globally or specifically in the prefrontal cortex, and determine whether this rescues the behavioral impairments.

We agree that in principle such a ‘rescue’ manipulation is a worthy long term aim, however with the current experimental tools and state of knowledge this would be extremely challenging for a number of compelling reasons. First, to our knowledge, whilst drug compounds do exist that have been shown to stimulate myelination in-vivo by enhancing oligodendrocyte proliferation (in the context of putative MS therapies), and one could speculate that they could be used, in theory, to alleviate the oligodendrocyte abnormalities and myelin thinning we observe, the current drugs are completely non-specific in their action. Furthermore, it is completely unknown how these drugs, most of which are thought to work via positive neurotrophic and/or anti-inflammatory effects, will interact with the specific abnormalities provoked by *Cyfp1* haploinsufficiency. It may be for example that what we need to ‘rescue’ is the myelin basic protein mal-translocation component of the phenotype or not. Basically we need

much more information on the phenotypes so that we can optimally target any rescue be that prosecuted using drugs or genetic approaches. Our work makes an important contribution to elaborating *Cyfp1* phenotypes using an entirely novel CRISPR/Cas9-engineered rat line and comprehensive cellular and behavioural analyses. Moreover, in highlighting white matter our findings will stimulate a focus on changes in brain connectivity as a mechanism underlying the greatly increased incidence of psychopathology seen in individuals with heterozygous deletions at the h15q11.2 interval.

4. In the reversal learning task in Figure 4d, how do the authors explain that some *Cyfp1*+/- rats were completely unable to learn the reversal, but those that did performed like normal WT rats? Is it possible that there are additional effects involved, e.g. off-target effects of the CRISPR strategy (despite the described testing for 10 potential OT sites)? Does the performance in the reversal learning task correlate with the reduction in myelination, possibly indicating a causal link?

The observation that a proportion of the mutant rats were able to perform the reversal task is an interesting point and one which we have ourselves thought a lot about. The practical reason we were not in a position to address this issue, as the reviewer indicates by correlating degrees of DTI/myelin changes with behavioural performance was that throughout the experiments we were careful to avoid confounds due to the experience of the animal. In this case we were concerned that the behavioural tasks themselves might impact on the brain measures and give rise to effects that were in addition to, and/or interacted with, the effects of *Cyfp1* haploinsufficiency. Similar considerations dictated our use of completely separate cohorts of subjects for all of the different analyses; DTI and electron microscopy as well as the two main behavioural tasks, reversal learning and associative mismatch. With respect to the two behavioural tasks it is of interest to note that, in contrast to reversal learning there were uniform group effects in the mismatch task. A priori this would in general tend to go against the idea, as suggested by this reviewer, of behavioural consequences arising from off-target effects (OTE). However we recognise the importance of having confidence that OTE do not contribute to phenotypes, especially in the light of recent analysis showing that Cas9-mediated off-target effects are relatively common (Anderson et al., 2018) which found that 23% of 81-genome-editing projects exhibited OTEs across mouse and rat models. As recommended by Anderson et al., 2018 what is needed to validate the specificity of CRISPR-Cas9 gene editing is a thorough assessment of sgRNA design (as detailed currently in the Supplementary Methods (Assessment of potential off-target effects), pages 9 to 13), in-silico prediction of potential OT sites and importantly biological assays, that together allow confidence that any OT engineered gene changes fall well below the background mutation frequency that can occur spontaneously.

To address this issue here we now show an expanded analysis of potential OT sites in Table 1 below and have now included this in the supplementary data Table I. It can be seen that the sgRNA guide designed against rat *Cyfp1* has a MIT specificity score (Hsu et al., 2013) of 100% against the *Cyfp1* gene and an overall score of 73.3% when factored against 49 OT sites. Previously in authoritative reviews it has been considered that a cut-off score of 66%, and above is sufficient to effectively eliminate the likelihood of OTE (Hsu et al., 2013) and this proposition has recently been confirmed by others (Anderson et al., 2018). Further, it has previously been shown that 2 mismatches (MM)

– in concatenated or interspaced form - reduce Cas9 cleavage activity to low levels and can be further reduced to negligible levels if they occur within close proximity (12bp) to the PAM region (Hsu et al., 2013). Moreover, 3 concatenated mismatches, were sufficient to eliminate detectable cleavage in the majority of loci tested, and more so, if these were interspaced and/or proximal to PAM. Of relevance to our situation, closer inspection of our list of Top 10 OT sites in Table 2, which we have again added to the supplementary data Table II, shows that all have a minimum of 3 MMs and sites 1 and 3 both possess a MM in close proximity to the PAM.

Having made the above arguments we also recognise that there has been recent discussion in the gene editing field as to the extent to which in-silico predicted OT sites (MIT website, Benchling) reflect true biological OT sites of Cas9-mediated activity. To some extent, this has been addressed experimentally by (Anderson et al., 2018), whereby 30 robust OT sites were identified by two sequencing methodologies (whole-genome sequencing and TEG-sequencing) and found that 25 of these, i.e. 83%, were predicted using in silico methodologies. Therefore, we can be reasonably confident that the Top 10 OTE table generated for the *Cyfp1* sgRNA is reflective of bone-fide biological OT sites. Critically though, we also showed in the case of our *Cyfp1* rat model that all of the Top 10 predicted OT sites were negative using the Surveyor Assay, i.e. no OTE were detected when we explicitly tested for them using wet methods. If OTE can be excluded this still leaves the question of what might explain why some mutants were able to perform the reversal learning task. As the rat line was on an out-bred background there is the possibility of modifying genes influencing the effects of *Cyfp1* haploinsufficiency, and here it is interesting to note that psychiatric and behavioural phenotypes are seldom expressed uniformly in people with the 15q11.2 deletion.

Table 1 - MIT predicted off target sites for de Cyfip1 sgRNA in-silico.

Numbered Ots	MIT specificity scores - using Benchling (previously MIT website)			Gene	Overall specificity score 73.3 %
	Sequence	PAM	Score		
	GGCAGATCCACAATCCATCC	AGG	100	Cyfip1 (ENSRNOG00000011945)	chr1:+114295724
OT1	GGAAAAATCCACCATCCATCC	TGG	1.4		chr2:-95793954
OT2	GCCAAATCCACAATCCATGC	CAG	1.3		chr14:+5813530
OT3	GTATCATCCACAATCCATCC	AAG	1.3		chr4:+109037870
OT4	CTCAGAAACACAATCCATCC	CAG	1		chr3:+90820151
OT5	GGTAGCCCCACAATCCATCC	CAG	1		chr8:+90295695
OT6	TGTACAGCCACAATCCATCC	AAG	1		chr8:+45237516
OT7	GTCCGAACCCCAATCCATCC	TAG	1		chr2:-85528916
OT8	TGGAAATCCACAATCCATCT	CAG	0.9		chr2:+223864733
OT9	AGCAGATCCAGATCCATCC	AGG	0.7		chr4:+72665650
OT10	AGCATATACACAATCCATCC	CAG	0.7		chr7:-113365387
OT11	TGCATACCCAGAATCCATCC	TAG	0.6		chr6:-128916162
OT12	AGCATAACCAGAATCCATCC	GGG	0.6		chr11:+79708047
OT13	GCTAGATCCTCAGTCCATCC	TGG	0.6		chr2:-172781048
OT14	AGCAGATGCCCAATCCATCC	GAG	0.6		chr18:-63836841
OT15	GACAAAGCCAAAATCCATCC	AGG	0.6		chr3:+28891768
OT16	GTCAGAGACAGAATCCATCC	AAG	0.6		chr4:-170865117
OT17	TGCACAGCCACAATCCATCC	AGG	0.5		chr14:+13356994
OT18	AGCACTCCACCATCCATCC	AGG	0.5		chr4:-180525716
OT19	AACTGATCCACAATCCATCC	CAG	0.5		chr8:-8347020
OT20	CACAGATCTACAATCCATCC	CAG	0.5		chr3:-176292068
OT21	GGCAGATCCAGAATCCATCC	CGG	0.5		chr6:-43384274
OT22	TGTAGATCCTCAATACATCC	AGG	0.5		chr7:+67511010
OT23	TCCAGATCCAGAATCCATCC	TAG	0.4		chr11:-36121578
OT24	GGCAGGTTCTGAATCCATCC	CAG	0.4		chr8:+24534839
OT25	GACACACCCACACTCCATCC	AGG	0.4		chr19:-55314147
OT26	AGCAGAGTCACAATCCATCC	AAG	0.4		chr2:-261967831
OT27	GGCAGCTGCGCAATCCATCA	TGG	0.4		chr3:-173954380
OT28	GGCAAATTCATGATCCATCC	AAG	0.4		chr10:+8354081
OT29	GGCATTGACAATCCATCC	CAG	0.4		chr2:-126822158
OT30	GGCGGACCCCAAGTCCATCC	AGG	0.4	Cyfip2 (ENSRNOG00000006557)	chr10:-31364956
OT31	TGCACATCCGCAATCCACCC	CAG	0.4		chr10:-92422723
OT32	TGCAGATACATATCCATCC	CAG	0.4		chr15:+1789082
OT33	GGAATGTCCACAGTCCATCC	AGG	0.4		chr18:-31761206
OT34	GGAATTTCCACATCCATCC	TGG	0.4		chr15:-61472675
OT35	AGCACATCCACTCTCCATCC	CAG	0.3		chr19:-30859164
OT36	GATAGATCCACTGTCCATCC	TGG	0.3		chr5:+63159170
OT37	AGCAGACTCACAATCCATCC	TAG	0.3		chr2:-85469475
OT38	ACCAGATCCACACTCTCCATCC	CAG	0.3		chr5:+135998305
OT39	GGCACATCCACAATCCAGCA	AAG	0.3		chr1:+146539887
OT40	TTGAGATCCACAATCTATCC	TAG	0.3		chr2:+103959476
OT41	TGCAGAACCACTATCCATCT	CGG	0.3		chrX:-63694162
OT42	AGCAGCTCCAGAATCCATCA	TGG	0.3		chr1:-146566385
OT43	TGCAGATCTAAAATCCATCT	GGG	0.3		chrX:-143494363
OT44	GCCAGCTCCAAAATCCATCT	AAG	0.3		chr3:+127954078
OT45	TGCAGTTCAGGATCCATCC	AAG	0.3	ENSRNOG00000020103	chr18:+30285706
OT46	GTCAGAACCAGAATCCCTCC	TGG	0.3		chr5:+75274315
OT47	GAAAGATCCATAATCCATCC	TGG	0.3		chr2:+253045233
OT48	TGCAAATCCACAGTCCATCC	AGG	0.3		chr5:-85205106
OT49	GCCAGGGCCACAATCCATCA	AGG	0.3	Grm3 (ENSRNOG00000005519)	chr4:-21535579

Table 2 - Top 10 predicted off target sites for the Cyfip1 sgRNA with a breakdown of mismatches and individual MIT scores; these were tested experimentally by the Surveyor Assay and shown to be negative.

	Sequence	Genomic Coordinates	Mismatches	MM proximal to PAM (12bp core)	Interspaced MM	MIT score
Injected US sgRNA sequence	GGCAGATCCACAATCCATCCAGG	Chr1:+114295729	0	0	0	100
OT Site 1	GGAAAATCCACCATCCATCCTGG	chr2:-115534496	3	1	3	1.4
OT Site 2	GTATCATCCACAATCCATCCAAG	chr4:+173741171	4	0	0	1.3
OT Site 3	GCCAATCCACAATCCATGCCAG	chr14:+5796700	3	1	3	1.3
OT Site 4	CTCAGAAAACAATCCATCCAG	chr3:+97482256	4	0	2	1
OT Site 5	TGTACAGCCACAATCCATCCAAG	chr8:+43715985	4	0	4	1
OT Site 6	GTCCGAACCCCAATCCATCCTAG	chr2:-105200718	4	1	4	1
OT Site 7	TGGAATCCACAATCCATCCTAG	chr2:+241911684	4	1	4	0.9
OT Site 8	AGCAGATCCAGATCCATCCAGG	chr4:+137367617	3	2	3	0.7
OT Site 9	TGCATACCCAGAATCCATCCTAG	chr6:-138113252	4	1	4	0.6
OT Site10	AGCATACCCAGAATCCATCCGGG	chr11:-81022173	4	1	4	0.6

Minor comment:

Figure 2a: Are the representative images for WT and Cyfip1^{+/-} on the same scale? The axons appear much larger in diameter in the WT than in the Cyfip1^{+/-} in this example, yet according to the quantification in Figure 2d there is no difference.

The images illustrated in Figure 2 are in the same scale, the apparent differences referred to are simply due to the presence of a few larger axons in the WT photomicrograph, this is a single picture containing approximately 50 axons whereas the overall quantification assayed around 14,000 axons.

References:

- Anderson, K.R., Haeussler, M., Watanabe, C., Janakiraman, V., Lund, J., Modrusan, Z., Stinson, J., Bei, Q., Buechler, A., Yu, C., et al. (2018). CRISPR off-target analysis in genetically engineered rats and mice. *Nat. Methods* 15, 512–514.
- Compston, A., and Coles, A. (2008). Multiple sclerosis. *Lancet Lond. Engl.* 372, 1502–1517.
- Feldman, D.E. (2012). The spike-timing dependence of plasticity. *Neuron* 75, 556–571.
- Fields, R.D. (2008). White matter in learning, cognition and psychiatric disorders. *Trends Neurosci.* 31, 361–370.
- Hsu, P.D., Scott, D.A., Weinstein, J.A., Ran, F.A., Konermann, S., Agarwala, V., Li, Y., Fine, E.J., Wu, X., Shalem, O., et al. (2013). DNA targeting specificity of RNA-guided Cas9 nucleases. *Nat. Biotechnol.* 31, 827–832.
- Huang, B., Wei, W., Wang, G., Gaertig, M.A., Feng, Y., Wang, W., Li, X.-J., and Li, S. (2015). Mutant huntingtin downregulates myelin regulatory factor-mediated myelin gene expression and affects mature oligodendrocytes. *Neuron* 85, 1212–1226.

Lee, Y., Morrison, B.M., Li, Y., Lengacher, S., Farah, M.H., Hoffman, P.N., Liu, Y., Tsingalia, A., Jin, L., Zhang, P.-W., et al. (2012). Oligodendroglia metabolically support axons and contribute to neurodegeneration. *Nature* 487, 443–448.

Markram, H., Gerstner, W., and Sjöström, P.J. (2012). Spike-Timing-Dependent Plasticity: A Comprehensive Overview. *Front. Synaptic Neurosci.* 4.

Mighdoll, M.I., Tao, R., Kleinman, J.E., and Hyde, T.M. (2015). Myelin, myelin-related disorders, and psychosis. *Schizophr. Res.* 161, 85–93.

Miyata, S., Hattori, T., Shimizu, S., Ito, A., and Tohyama, M. (2015). Disturbance of oligodendrocyte function plays a key role in the pathogenesis of schizophrenia and major depressive disorder. *BioMed Res. Int.* 2015, 492367.

Olmos-Serrano, J.L., Kang, H.J., Tyler, W.A., Silbereis, J.C., Cheng, F., Zhu, Y., Pletikos, M., Jankovic-Rapan, L., Cramer, N.P., Galdzicki, Z., et al. (2016). Down Syndrome Developmental Brain Transcriptome Reveals Defective Oligodendrocyte Differentiation and Myelination. *Neuron* 89, 1208–1222.

Philips, T., and Rothstein, J.D. (2014). Glial cells in amyotrophic lateral sclerosis. *Exp. Neurol.* 262 Pt B, 111–120.

Pouwels, P.J.W., Vanderver, A., Bernard, G., Wolf, N.I., Dreha-Kulczewski, S.F., Deoni, S.C.L., Bertini, E., Kohlschütter, A., Richardson, W., Ffrench-Constant, C., et al. (2014). Hypomyelinating leukodystrophies: translational research progress and prospects. *Ann. Neurol.* 76, 5–19.

.....

Reviewer #2

We thank this reviewer for their positive remarks and opinion of the work. The reviewer raised some issues which we reply to below.

Silva et al.

This paper is extremely well written and presented. The work is thorough and well thought out. The authors present a novel heterozygous *Cyfp1* rat model. They examined the rat with DTI to look for white matter microstructure differences as well as electron microscopy to investigate their DTI results and immunohistochemistry to further characterize the differences. I appreciate the extra care that was put into this study in order to confirm the FA findings with EM. I furthermore appreciate the statistical examination that was performed. My comments are relatively minor as I found the paper to be quite good as is.

1) The last sentence of the abstract made me expect some resting-state fMRI looking at functional connectivity or circuitry the way it is worded. Perhaps some mention of the behavioural tasks.

We have removed the work ‘functional from the last line of the Abstract to avoid any confusion. However we cannot add any detail of the behavioural tasks used as this would take us over the strict work limit for the Abstract.

2) A b-value shell of 1000 s/mm² seems a little small for the rat. Why was this chosen?

A b-value shell of 1000 s/mm² is the typical value used in in-vivo rat and mice imaging. This value was chosen to take into account signal-to-noise ratio and data quality, since higher b-values, together with movement associated with in-vivo imaging, could introduce artefacts and therefore compromise data quality. Usually, higher b-values are applied in DTI studies using ex-vivo animal preparations, where movement is excluded.

3) Did you test for any behavioural/brain correlations in the touch screen task and the FA measurements? Or were they different cohorts?

a. I’m curious if you can see any notable FA differences across the *Cyfp1* (or WT for that matter) based on their performance in the tasks.

The practical reason we were not in a position to address this issue was that throughout the experiments we were careful to avoid confounds due to the experience of the animal. In this case we were concerned that the behavioural tasks themselves might impact on the brain measures in addition to, and/or interacting with, the effects of *Cyfp1* haploinsufficiency. These considerations dictated our use of completely separate cohorts of subjects for all of the different analyses; DTI, electron microscopy and the two main behavioural tasks, reversal learning and associative mismatch, the one exception was for the DTI and immunofluorescence measures which used the same cohort of animals.

4) Did you don any correlations between the immunohisto and the FA measurements since they were on the same mice?

As above, this analysis was possible due to the same animals being used but when we carried out the analysis there was no significant correlation.

.....
Reviewer #3

We thank the reviewer for their positive remarks about the novelty and innovation of the study and their critique of the behavioural studies the specific points of which we reply to below. We have also responded to the comments and insights of this reviewer by making changes to the behavioural discussions throughout the main text in order to aid clarity for the reader.

Silva and colleagues have generated a novel innovative genetic rat model with *cyfp1* deficiency, an important protein for actin dynamics with links to schizophrenia and autism spectrum disorder. They used this novel rat model to assess a clinically relevant level of low dose of *Cyfp1* over other mouse models that KO. They present data on alterations in white matter microstructure by DTI, pronounced in the corpus callosum and external capsule. They also performed finer microscopy on these outcomes. Finally, they used advanced touchscreen cognitive behavioral testing to illustrate cognitive dysfunction on specific executive learning tasks.

Behavioral Critique:

There were a few major behavioral critiques.

It was disappointing that the same animals were not used for behavior and imaging, especially since the authors performed in vivo imaging. This was an obvious missed opportunity. It would seem a subset of the behavioral animals could have been tested for within subject correlations. In differing animals, how do we know the “level of circuitry impairment” would yield “level of behavioral impairment”

This is a good point but there were compelling reasons why we did not determine imaging measures in tandem with behavioural measure and therefore, as the reviewer indicates be in a position to correlate imaging and behavioural phenotypes. The reason was that we were careful to avoid potential confounds related to the experience of the animal, in this case we were concerned that the behavioural tasks themselves might impact on the DTI measures, in addition to, and/or possibly interacting with, the effects of *Cyfp1* haploinsufficiency. Similar considerations dictated our use of completely separate cohorts of subjects for all of the different analyses; DTI and electron microscopy as well as the two main behavioural tasks, reversal learning and associative mismatch. The one exception was for the DTI and immunofluorescence as applied to oligodendrocytes which used the same cohort of animals, however when we ran the correlation there was no reliable systematic relationship between variability in this particular histological measure and the differences in FA we observed on the basis of genotype.

The nomenclature is off. The pairwise reversal learning task is just that “reversal learning”, it is not go/no-go nor inhibition.

This text is incorrect and misleading “Successful reversal is demanding on attention and response inhibition, and can be viewed as a type of “go”– “no”. The field would not agree on this statement.

Reversal in the touchscreen or an operant box is mediated by the orbitofrontal cortex and OFC mediated reversal is specifically NOT response inhibition.

Schoenbaum G, Nugent SL, Saddoris MP, Setlow B (2002)

Orbitofrontal lesions in rats impair reversal but not acquisition of go, no-go odor discriminations. *NeuroReport* 13:885–890.

Schoenbaum G, Roesch MR, Stalnaker TA, Takahashi YK (2009b) A new perspective on the role of the orbitofrontal cortex in adaptive behaviour. *Nat Rev Neurosci* 10:885–892.

The neural basis of reversal learning: an updated perspective

A Izquierdo, JL Brigman, AK Radke, PH Rudebeck, A Holmes

Neuroscience 345, 12-26

We agree with the reviewer and have removed the relevant text and in particular the erroneous references to ‘go/no go’ and ‘response inhibition’ and instead describe these tasks as reflecting the more general process of behavioural flexibility, which as stated within the manuscript is known to involve key areas such as the frontal cortex (including the OFC) and the striatum. In response to these points we have also made changes, highlighted throughout the main text, to aid clarity in regard to the behavioural work.

Figure 4C should be illustrated as the field standard of early, mid, late reversal, as shown in Marquardt et al., 2017, 2018 and Brigman et al., Nature Neuro 2013.

We deliberately reported the very first few sessions of reversal learning as this was when the impact of the change in contingencies would be most evident, and was in line with previous studies who have looked at reversal performance on a session by session basis (Ragozzino et al., 2009; Varga et al., 2014; Weisenhaus et al., 2010; Whitehouse et al., 2017). This was important as it enabled us to demonstrate that, whilst there were no genotype differences in the initial response to reversal (session 1) the performance of the *Cyfp1*^{+/-} rats in this immediate post-reversal phase was more influenced by the previous contingencies (and corresponding pre-potent response) than WT rats irrespective of whether these animals eventually went on to attain any arbitrary reversal criterion, suggesting that *Cyfp1*^{+/-} were significantly slower to adapt to the changes in response contingencies (sessions 2-4, figure 4c (Figure 5c in the revised manuscript)). Hence, presenting the data in the way suggested (i.e. performance on the first session of reversal, then to 50% correct, and criterion [e.g. 80% correct]) would have failed to reveal this effect on behavioural flexibility as some of the *Cyfp1*^{+/-} rats were able to successfully overcome this early perseverative effect and go on to complete reversal, though significantly fewer than WT. This raises the question of why some of the mutant rats were able to eventually complete reversal. Given we have argued elsewhere that this is very unlikely to be due to off-target effects of the genetic editing methods used to create the rat model, the most parsimonious explanation is that, as Long Evans are an out-bred line there may be modifying genes influencing the effects of *Cyfp1* haploinsufficiency. In this regard it is of interest to note that psychiatric and behavioural phenotypes, including cognitive phenotype, are also not expressed uniformly in people with the 15q11.2 deletion.

Figure 4D shows that only 50% of WT completed the reversal to 80%, this is highly unusual especially for LE background strain. In our experience and prior literature, all animals plus or minus 1 complete, however it could be that they did not continue testing to go out to 30 days or more. This makes me think 4D may be an artifact of the shortened testing period.

The task in our hands was consistent with previous findings using touch-screen methods in rat in that subjects that did complete the reversal task did so in a similar number of trials and sessions to those reported by the pioneers of this widely adopted approach (Horner et al., 2013). With regard to the specific point about the animals that failed to complete reversal, both WT and *Cyfp1*^{+/-} were given extensive opportunity to reach criterion, receiving on average 29 sessions training. In contrast, those animals that did complete the reversal phases did so in an average of 20 sessions, so we do not believe that these data are an artefact of reduced exposure to the contingencies. In addition, when we terminated the experiment for these animals they were on average performing at 45% correct, and so were not close to reaching criterion. Extending this training might result in more animals reaching criterion and so may alter the data for sessions to criterion in Figure 5e but crucially it would not influence the main finding that *Cyfp1*^{+/-} rats were slower to adapt to the reversal contingencies in the early reversal sessions. Moreover, the finding of reduced behavioural flexibility in *Cyfp1*^{+/-}, as assayed in the reversal learning task, was supported by the data obtained independently with a separate cohort of animals in the associative mismatch task.

The second behavioral assay is not well described by literature. Could the authors provide more referencing? If this is the first description, they should report how they validated this assay as measuring the complex processes that underlie it (lesion or pharm or dreadd blockage of signaling).

The mismatch task we used in the behavioural work has been described extensively in at least 4 previous publications (Honey and Good, 2000a, 2000b; Honey et al., 1998, 2010), including assessing the effects of task variables and brain lesions, and has been cited extensively by other researchers. Importantly, the task allowed us to examine behavioural flexibility in the *Cyfp1^{+/-}* model using changes in contingencies related to sensory events (auditory-visual pairings). This enabled us to generalise difficulties in behavioural flexibility beyond the reversal and reward learning circuitry that is engaged by the touchscreen task. We have now added further description and additional references for this second assay within the main text.

Why is the N so substantially larger for the second task?

The Ns were chosen on the basis of previous experience with rats in the two tasks and the literature, and were sufficiently powered to reveal reliable differences in behaviour. The reversal learning data reflects the passage of the animals through the task, 1 *Cyfp1^{+/-}* animal did not reach the criterion performance for the initial learning of the visual discrimination, and 1 WT did not start the reversal learning, and so were not taken through the rest of the task elements. All the animals progressed through the whole of the associative mismatch task.

Where are the control parameters that Lafayette provides you for motor? Trial number?
Latency to collect reward (which tells us about motivation)

Again, a good point but we have provided trial and session numbers for each of the phases in the supplementary material. These clearly show that *Cyfp1^{+/-}* rats have no difficulty in completing pre-training phases (habituation and must touch), nor are there any differences in acquisition of the initial visual discrimination task. There were also no differences in latency to collect rewards in these animals (unpublished data) during these phases. Therefore, any genotype effects are likely driven by the changes in task requirements during the reversal phase, and so unlikely to be driven by differences in motor behaviour or motivation.

What do the authors think about that the circuitry that they've disrupted is not classically associated with these behaviors? Discussion should be added to expand their opinion of this and acknowledge the breadth of literature on the key neural correlate of the OFC in the reversal task.

Whilst there is some previous research directly examining the role of the corpus callosum in behavioural flexibility, the literature is not extensive. The corpus callosum, however, is known to be important in the functional integrity of brain regions (i.e. orbitofrontal cortex, prefrontal cortex and striatum (Haber and Behrens, 2014; Ozalay et al., 2013; Putnam et al., 2008) that are classically linked with behavioural flexibility so it is not unlikely that changes to the integrity of the corpus callosum will impact performance on such tasks. We have made changes in the discussion to highlight this and to mention the key role of the OFC in reversal learning.

References

- Haber, S.N., and Behrens, T.E.J. (2014). The Neural Network Underlying Incentive-Based Learning: Implications for Interpreting Circuit Disruptions in Psychiatric Disorders. *Neuron* 83, 1019–1039.
- Honey, R.C., and Good, M. (2000a). Associative components of recognition memory. *Curr. Opin. Neurobiol.* 10, 200–204.
- Honey, R.C., and Good, M. (2000b). Associative modulation of the orienting response: Distinct effects revealed by hippocampal lesions. *J. Exp. Psychol. Anim. Behav. Process.* 26, 3–14.
- Honey, R.C., Watt, A., and Good, M. (1998). Hippocampal Lesions Disrupt an Associative Mismatch Process. *J. Neurosci.* 18, 2226–2230.
- Honey, R.C., Iordanova, M.D., and Good, M.A. (2010). Latent inhibition and habituation: evaluation of an associative analysis. In *Latent Inhibition: Cognition, Neuroscience and Applications to Schizophrenia*, R. Lubow, and I. Weiner, eds. (Cambridge: Cambridge University Press), pp. 163–182.
- Horner, A.E., Heath, C.J., Hvoslef-Eide, M., Kent, B.A., Kim, C.H., Nilsson, S.R.O., Alsiö, J., Oomen, C.A., Holmes, A., Saksida, L.M., et al. (2013). The touchscreen operant platform for testing learning and memory in rats and mice. *Nat. Protoc.* 8, 1961–1984.
- Ozalay, O., Calli, C., Kitis, O., Cagdas Eker, M., Donat Eker, O., Ozan, E., Coburn, K., and Saffet Gonul, A. (2013). The relationship between the anterior corpus callosum size and prefrontal cortex volume in drug-free depressed patients. *J. Affect. Disord.* 146, 281–285.
- Putnam, M.C., Wig, G.S., Grafton, S.T., Kelley, W.M., and Gazzaniga, M.S. (2008). Structural Organization of the Corpus Callosum Predicts the Extent and Impact of Cortical Activity in the Nondominant Hemisphere. *J. Neurosci.* 28, 2912–2918.
- Ragozzino, M.E., Mohler, E.G., Prior, M., Palencia, C.A., and Rozman, S. (2009). Acetylcholine activity in selective striatal regions supports behavioral flexibility. *Neurobiol. Learn. Mem.* 91, 13–22.
- Varga, A.W., Kang, M., Ramesh, P.V., and Klann, E. (2014). Effects of acute sleep deprivation on motor and reversal learning in mice. *Neurobiol. Learn. Mem.* 114, 217–222.
- Weisenhaus, M., Allen, M.L., Yang, L., Lu, Y., Nichols, C.B., Su, T., Hell, J.W., and McKnight, G.S. (2010). Mutations in AKAP5 Disrupt Dendritic Signaling Complexes and Lead to Electrophysiological and Behavioral Phenotypes in Mice. *PLOS ONE* 5, e10325.
- Whitehouse, C.M., Curry-Pochy, L.S., Shafer, R., Rudy, J., and Lewis, M.H. (2017). Reversal learning in C58 mice: Modeling higher order repetitive behavior. *Behav. Brain Res.* 332, 372–378.

REVIEWERS' COMMENTS:

Reviewer #1 (Remarks to the Author):

The authors have addressed all of my comments, and the addition of the cell culture data provides interesting insights into potential mechanisms linking CYFIP1 to myelination.

Reviewer #2 (Remarks to the Author):

The authors have address all of my concerns

Reviewer #3 (Remarks to the Author):

The majority of my behavioral concerns were addressed in the revised manuscript.
Minor comments

1. Animals that are removed from behavioral testing that change final sample size should be stated in the results text section.
2. A citation for the confounds of the behavioral experiences and DTI would be a nice addition.
3. The Honey et al., citations in the rebuttal letter use latent inhibition, are the authors suggesting that their associative mismatch is equal to latent inhibition or something novel, this just needs a little bit more clarity.